# Predictive patterning via solid-state dewetting of transferred single-crystal films

Seungjin Ju [1,5], Sangsun Lee[1,5], Donghwan Kim[1], Hyunsik Kim[1], Jaeho Lee [1,2], Jeong-Hwan Lee [1,2], Wi Hyoung Lee [3], Olivier Pierre-Louis [4] & Jongpil Ye [1] ✉

Designing and exploiting the dewetting of single-crystal films to create specific patterns for fabricating functional structures requires improved predictability and extensibility. In this study, we demonstrate that templated solid-state dewetting of single-crystal films can be guided to form regular patterns on arbitrary surfaces. This is achieved by adopting multiscale calculation schemes and implementing the dewetting results of single-crystal Pd(100) films transferred onto amorphous $SiO_2$ substrates. The anisotropies of Pd surface energy and gas adsorption strength lead to <001> and <011> in-plane facets, favoring the latter as oxygen adsorption increases or the hydrogen flow rate is sufficiently high. This leads to anisotropic dewetting patterns whose geometric characteristics depend significantly on initial crystallographic alignment and annealing ambient. A combination of computational and experimental methods is used to design and guide dewetting to create electrode patterns with submicron channels for thin-film transistors, demonstrating their feasibility for fabricating functional structures on various substrates.

Solid-state dewetting of single-crystal films occurs through the nucleation and growth of holes whose shapes are consistent with the crystallographic symmetries of the film surfaces, leading to morphologies with highly regular characteristics[1–5]. This has motivated research efforts to exploit the morphological evolution as a novel patterning process for creating regular structures. In most of these studies, the films were lithographically pre-patterned to apply geometric confinement to their crystallographically constrained morphological evolution, resulting in substantially enhanced regularity of the resulting structures[6–10]. The regular structures have sub-lithographic feature sizes and geometries with increased complexity owing to the various morphological instabilities involved in their formation and their dependence on the surface energy anisotropy of the film material. This renders the patterning process applicable to fabricating functional structures that cannot be achieved via conventional dewetting processes.

The potential of the dewetting process as a patterning method has not been sufficiently realized, for two main reasons. First, the choice of substrate is highly limited because the dewetting process requires crystallographic commensurability and chemical inertness between the film and the substrate materials, which are necessary for epitaxial growth and high-temperature annealing, respectively. For example, only single-crystal oxide substrates, such as MgO and $Al_2O_3$, have been used in studies on the dewetting of single-crystal metal films, limiting the applications of dewetted single-crystal metallic structures[6,11,12]. Furthermore, such single-crystal substrates are expensive, and their use for large-area patterning is not affordable. Second, despite significant progress, up-to-date reported simulation methods face challenges in precisely guiding the process conditions for creating the desired patterns through the dewetting of single-crystal films[11–15]. Simulation results based on a kinetic Monte Carlo (KMC) model captured the anisotropic nature of dewetting morphologies[15–17]; however, all KMC simulation parameters were empirically determined, and no theoretical framework

[1]Department of Materials Science and Engineering, Inha University, Incheon, Korea. [2]Program in Semiconductor convergence, Inha University, Incheon, Korea. [3]Department of Materials Science and Engineering, Konkuk University, Seoul, Korea. [4]Institut Lumière Matière, UMR5306 Université Lyon 1—CNRS, Villeurbanne, France. [5]These authors contributed equally: Seungjin Ju, Sangsun Lee. ✉e-mail: jpyecs@gmail.com; jpye@inha.ac.kr

was included to account for the chemical states of the film surfaces. These limitations make them unfeasible to consider the dependence of surface energy anisotropy on film materials and gas adsorption, which have been experimentally demonstrated to be critical in metal films.

Here, we demonstrate that these two shortcomings can be substantially improved through the transfer of large-area epitaxial metal films grown on Si substrates to target substrates and multiscale calculations, including density functional theory (DFT), molecular dynamics (MD), and the KMC methods. We present the formation of various regular patterns via solid-state dewetting of Pd(100) films transferred and patterned on oxidized, heavily doped Si wafers. The approach using film transfer removes the need for single-crystal oxide substrates and the associated process constraints that have limited the applicability of prior studies. The geometric characteristics of the dewetting patterns are shown to strongly depend on the initial crystallographic alignments of the patterns and the annealing ambient conditions. The morphological evolution during dewetting is well predicted using multiscale calculation schemes that adopt the surface energy anisotropy of the film materials and its changes with oxygen adsorption. The initial pattern layouts and annealing processes are designed according to the calculation results to form Pd line patterns on which Au films are lifted-off, leading to the formation of Au electrode patterns with submicron channels for back-gate indium gallium zinc oxide (IGZO) thin-film transistors (TFTs). These results demonstrate the feasibility of designing and optimizing dewetting processes to create specific functional structures on arbitrary surfaces.

## Results

### Ambient-dependent dewetting behaviors of transferred Pd(100) films

Selective wet etching of the Cu layer in a Pd/Cu stack epitaxially grown on an Si(100) wafer leads to a floating Pd layer on the etchant, which can be transferred onto a target substrate (see the schematic in Fig. 1a). As shown in Fig. 1b, Pd films were transferred and lithographically patterned over an area of a few $cm^2$ on oxidized Si wafers. The X-ray diffraction (XRD) and selected-area electron diffraction (SAED) measurements shown in Fig. 1c, d indicate that the transferred films were nearly single-crystal Pd(100) films. The single crystallinity of the transferred films was further supported by the in-plane alignment of holes with <011> edges, which were formed after the thermal annealing (see Fig. 1e).

Figure 2a, b shows the dewetting results of the patches lithographically patterned from the transferred Pd(100) films to be aligned along different crystallographic directions. These patches were annealed under two different annealing ambient conditions: 0.5 sccm $H_2$ and 50 sccm Ar in Fig. 2a; 2000 sccm Ar in Fig. 2b. These two ambient conditions prevent extensive oxidation during the annealing. As shown in the figures, the retraction of the external edges and the growth of internal circular holes in the patches resulted in the formation of ring patterns when the initial edge orientations were parallel to particular directions that correspond to the <011> or <001> directions. In contrast, patches aligned along other directions dewetted to form an array of islands or nonuniform ring patterns. This is because the retracting and thickening edges are faceted parallel to the two directions in which they are crystallographically constrained to expose low-energy facets such as (111) and (100) along their surfaces. Edges and lines exposing low-energy facets are stable against Rayleigh–Plateau-type instability[18,19], making ring patterns relatively stable[6,7]. An initial misalignment of patch edges from the specific directions leads to substantial width and thickness perturbation of the lines, resulting in the formation of particles via the Rayleigh–Plateau-type instability.

The annealing ambient conditions affected the in-plane orientation-dependent shapes of the dewetting patterns. As shown in Fig. 2a,

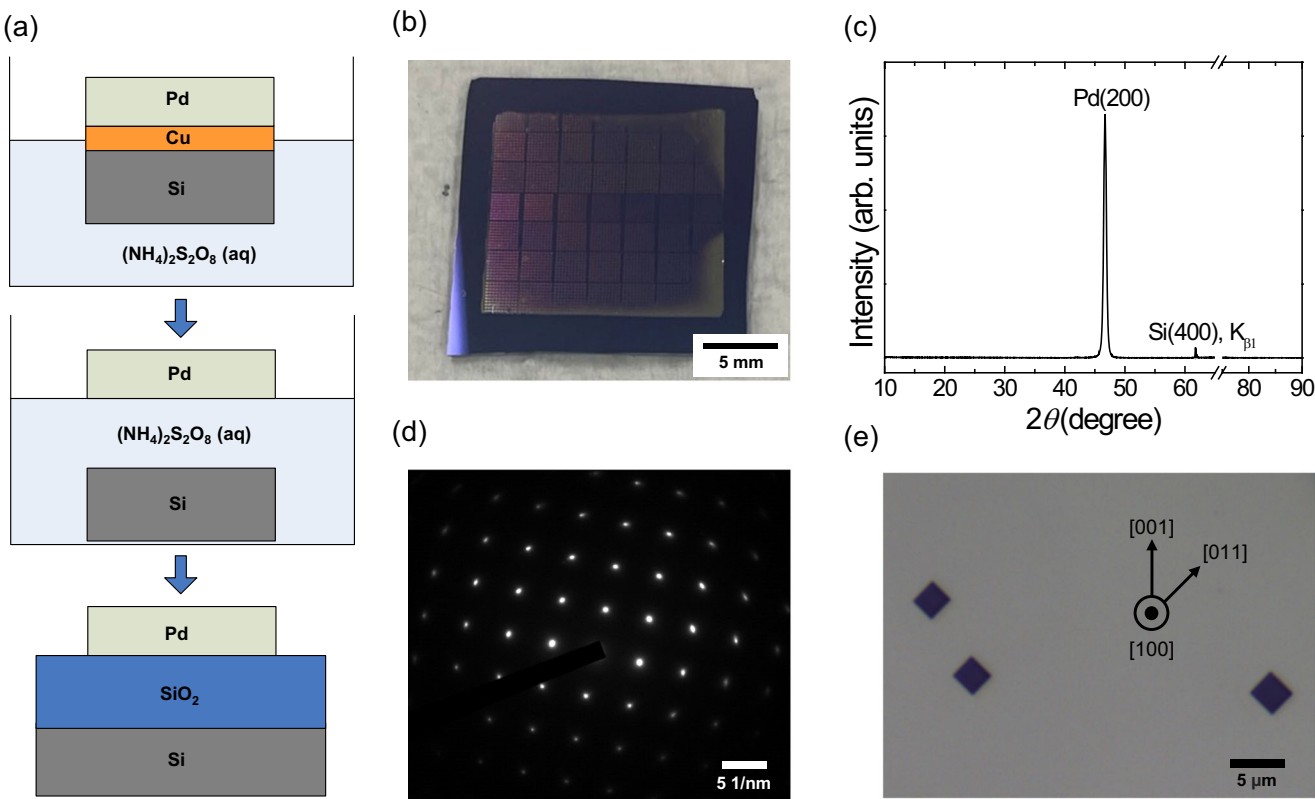

**Fig. 1 | Transfer of Pd(100) films on oxidized Si wafers and their characterization. a** Schematic illustrating the transfer process of a Pd film; **b** photograph after patterning; **c** $\theta$–$2\theta$ XRD measurement result; **d** SAED pattern of a 120-nm-thick transferred film. The zone axis of the SAED pattern is [100]. **e** OM image captured after the dewetting of a transferred film at 800 °C for 90 min under 2000 sccm Ar. Source data are provided as a Source Data file.

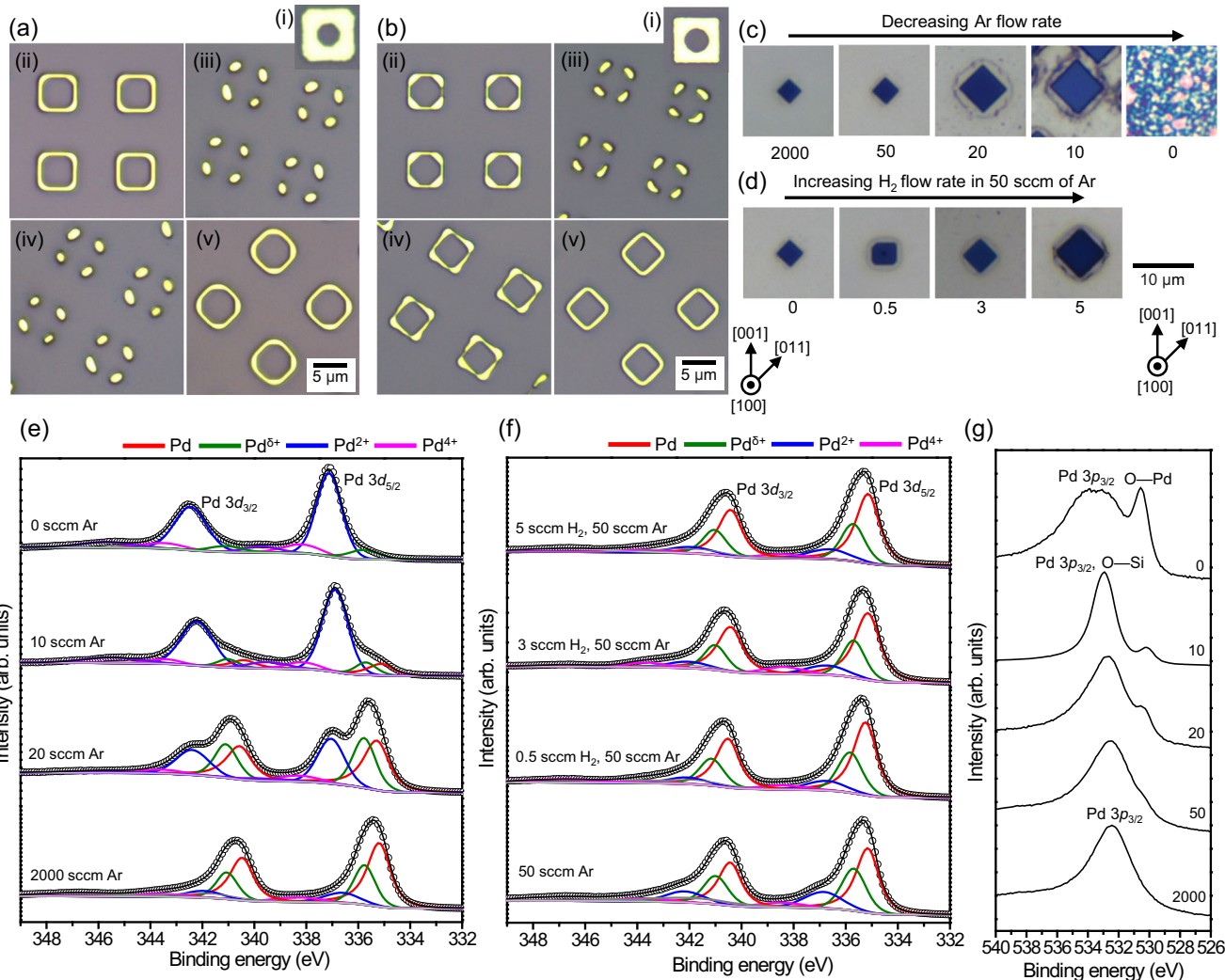

**Fig. 2 | Effects of the annealing ambient conditions and the crystallographic alignments of patches on the dewetting results of 120-nm-thick Pd(100) films.** **a**, **b** Optical microscopy (OM) images of patterns created via the dewetting of square patches with internal circular holes. Patches were annealed in **a** 0.5 sccm $H_2$ and 50 sccm Ar and **b** 2000 sccm Ar. The in-plane crystallographic orientations are indicated in the lower-right corner of (**b**). The initial shapes of patches are shown in (**a**(i)), (**b**(i)) (upper-right corner of (**a**, **b**)). For (ii), (iii), (iv), and (v) in (**a**, **b**), the edges of the initial patches were at angles of 0°, 15°, 30°, and 45° from the <001> directions, respectively. **c**, **d** OM images of holes formed during dewetting under different annealing ambient conditions. Ar flow rates were varied with no $H_2$ flow in (**c**), and $H_2$ flow rates were varied in 50 sccm Ar in (**d**). The numbers below each image in (**c**, **d**) indicate the flow rates of Ar and $H_2$, respectively. The in-plane crystallographic orientations are indicated in the bottom-right corner of (**d**). **e**, **f** Pd 3*d* XPS spectra for annealed Pd (100) films. The annealing ambient conditions for each spectrum are labeled in the figure. **g** O 1*s*–Pd 3*p* XPS spectra for Pd(100) films annealed at different Ar flow rates (numbers to the right of each spectrum in sccm). The films characterized in (**e**–**g**) correspond to those shown in (**c**, **d**). The annealing temperature and time were 800 °C and 90 min in all cases. Source data are provided as a Source Data file.

patches whose external edges were initially aligned along the <001> or <011> directions evolved into uniform ring patterns during the dewetting in 0.5 sccm $H_2$ and 50 sccm Ar. The squareness of the ring patterns is slightly greater when the initial alignment direction is <001>. In 2000 sccm Ar with no $H_2$, only the patches whose external edges were initially aligned along the <011> directions dewetted to form uniform ring patterns (see Fig. S1 in the Supplementary Information for enlarged images). Uniform ring patterns were observed when the external edge orientations corresponded to the orientations of the in-plane facets formed during the growth of the internal holes. Because the in-plane facet orientations depend on the annealing ambient conditions, the particular orientations where uniform ring patterns form differ under the two different ambient conditions, as shown in Fig. 2a, b.

Annealing ambient-dependent in-plane facet orientations were clearly observed in the shapes of the holes formed during the dewetting of continuous Pd(100) films. Figure 2c, d shows the holes formed

during the dewetting of the continuous Pd(100) films under various annealing ambient conditions. The in-plane facets of the holes were aligned along the <011> directions in 2000 sccm Ar, whereas primary <001> and secondary <011> in-plane facets were observed in 0.5 sccm $H_2$ and 50 sccm Ar. This is consistent with the dewetting results shown in Fig. 2a, b. The morphological details of the dewetting patterns varied with the initial patch geometry, which was affected by process fluctuations during photolithography or wet etching (see Fig. S2 in the Supplementary Information). Nevertheless, the key differences in the anisotropic dewetting behaviors between the two annealing ambient conditions were consistently observed across patches with different initial geometries.

### Effects of Pd surface energy anisotropy and oxygen adsorption
The effects of annealing ambient conditions on dewetting morphologies, including the shapes of dewetting patterns and growing holes,

have been attributed to gas adsorption-induced changes in the surface energy anisotropy and faceting of metals[20]. The in-plane facet orientations in the holes remained <011> throughout the reduction of Ar flow rate from 2000 sccm down to 10 sccm, at which the oxidation-induced change of the surface was clearly observed by optical microscopy (OM), as shown in Fig. 2c. This change could occur because reducing the Ar flow rate increased the adsorption of ambient oxygen. Conversely, increasing the $H_2$ flow rate to 0.5 sccm in 50 sccm Ar led to the formation of primary <001> in-plane facets, as described in the previous section. However, the relative lengths of these <001> facets gradually decreased with increasing $H_2$ flow rates of 0.5, 1, 1.5, and 2 sccm (see Fig. S3 in the Supplementary Information) and nearly disappeared at higher flow rates (3 and 5 sccm). Hydrogen is consumed in reducing surface oxides under annealing conditions in which its absence leads to oxidized surfaces. Excess hydrogen after reduction can be adsorbed on the surface or absorbed into the surface region. Hence, the amounts of oxygen and hydrogen adsorbates could depend on the hydrogen flow rates, which might be associated with the change of in-plane facets by the $H_2$ flow rate.

X-ray photoelectron spectroscopy (XPS) was performed to investigate the chemical states of the annealed Pd(100) film surfaces. Figure 2e, f show the Pd $3d$ spectra measured from the samples shown in Fig. 2c, d. As shown in the figures, reducing the Ar flow rate in the absence of $H_2$ increased the fractions of $Pd^{2+}$ and $Pd^{\delta+}$ species, which have been attributed to oxidized Pd in previous studies[21–23]. In particular, the fraction of $Pd^{2+}$, which is usually assigned as PdO[22,23], increased significantly. In the overlapping O $1s$–Pd $3p$ region (Fig. 2g), the peak intensity near 530 eV, which corresponds to oxygen adsorbates and surface oxides, also increased as the Ar flow rate decreased. This observation supports the assignment of the high-binding-energy components in the Pd $3d$ spectra to oxidized Pd[23,24].

Conversely, introducing 0.5 sccm $H_2$ into the 50 sccm Ar stream decreased the fractions of $Pd^{2+}$ (PdO) and $Pd^{\delta+}$ species while increasing the metallic Pd component (Fig. 2f), confirming the reducing effect of hydrogen. As shown earlier, the primary in-plane facet orientation was <011> in the absence of $H_2$ and changed to <001> at 0.5 sccm $H_2$ and 50 sccm Ar. The $Pd^{\delta+}$ peaks were shifted approximately by 0.5–0.6 eV to higher binding energies relative to the metallic Pd peaks. These shifts correspond to the reported values for oxygen-adsorption-induced shifts rather than those induced by hydrogen adsorption or absorption[23,24]. Hence, the abovementioned changes in the fraction of $Pd^{2+}$ and $Pd^{\delta+}$ in the Pd $3d$ spectra suggest that oxygen adsorption promotes the formation of <011> in-plane facets.

Adventitious carbon and sulfur were also detected in most Pd films used in this study (see Fig. S4 in the Supplementary Information for representative spectra). No correlation was found between these contaminants and the specific hole morphologies, but sulfur can also shift the Pd $3d$ spectra to higher binding energies[25], which may contribute to an increase in the $Pd^{\delta+}$ peak intensity. Hence, the $Pd^{2+}$ peaks, whose intensities were shown to correlate with those of surface oxide in the O $1s$ spectra, are more reliable indicators of the extent of oxidation.

The decrease in the fraction of $Pd^{2+}$ and $Pd^{\delta+}$ species were also observed in films annealed at higher $H_2$ flow rates as expected, as shown in Fig. 2f, indicating the role of $H_2$ in reducing the oxidized species. Although the results shown in Figs. 2c, d and S3 show that hydrogen has a critical effect on the reemergence of the dominant <011> in-plane facets, no distinct feature was consistently found in the XPS spectra of films annealed at higher $H_2$ flow rates. This may be due to the limitation of ex situ measurements, considering the films were exposed to ambient air prior to the measurements. In situ surface cleaning and measurements under precisely controlled partial pressures of hydrogen will be required to clarify the role of $H_2$ in the dewetting process.

As mentioned previously, the formation of low-index facets along the film surfaces plays an important role in determining the anisotropic shapes of the dewetting patterns and holes. Figure 3 shows the cross-sectional surface morphologies of the holes and patterns corresponding to those presented in Fig. 2. In all cases, rim morphologies were observed around the holes, indicating that the dewetting process was predominantly governed by surface diffusion. In the case of patterns formed at 2000 sccm Ar, longer dewetting line patterns were used to observe facet morphologies along the <001> directions (see the scanning electron micrograph inset of Fig. 3b), because uniform ring patterns did not form in those directions. As shown in Fig. 3a−d, the surfaces of the patterns aligned along the <011> and <001> directions have wider {111} and {100} facets, respectively, when annealed in 2000 sccm Ar than in 0.5 sccm $H_2$ and 50 sccm Ar. The morphological evolution of faceted surfaces can be modeled to be driven by a gradient in a weighted curvature, which is defined as follows: $\kappa = \frac{\sigma\Lambda}{L}$, where $\Lambda$ represents the facet length in the equilibrium crystal shape, $L$ represents the actual facet length, and $\sigma$ is the convexity factor (see Fig. S5 in the Supplementary Information)[26]. The wide facets indicate that surface energies at those orientations have prominent minima, implying that the $\Lambda$ values are large therein. The weighted curvature is maintained as very large, where the surface normal corresponds to those orientations until a wide facet forms and the value of $L$ becomes sufficiently large. Hence, the formation of a wide facet can significantly slow the surface diffusion flux from the edge front to the top film surface when the position of the facet is between the two locations, as in the case of the {111} facets in the patterns aligned along the <011> directions. When the facet orientation corresponds to the edge front, as in the case of the {100} facets in the patterns aligned along the <001> directions, a high-weighted curvature at the edge front can enhance the surface diffusion flux. This leads to slow edge retraction in the <011> directions and consequently causes the internal holes to be bounded by <011> in-plane facets during their growth. This explains the difference between the two annealing ambient conditions in terms of the direction in which uniform ring patterns form. The above description, based on the weighted curvature, can be related to the Winterbottom construction, which generalizes the classical Wulff construction to supported films by incorporating the film–substrate interfacial energy. Within this framework, the dewetting process can be interpreted as a morphological evolution toward the equilibrium shape defined by the Winterbottom construction. This evolution is driven by gradients in the weighted curvature defined as the surface divergence of the Cahn–Hoffman vector[14,27].

The cross-sectional surfaces along the thickened edges of the holes also exhibit facet morphologies consistent with those observed in the dewetting patterns. As shown in Fig. 3e−h, wide {111} or {311} facets were also formed along the thickened edges of the holes bound by <011> in-plane facets. In the case of holes bound by <001> in-plane facets, which were only observed under one annealing ambient condition (0.5 sccm $H_2$ and 50 sccm Ar), {110} facets were observed along the thickened edges. The contrasts shown in the scanning electron microscopy (SEM) images of the films annealed in $H_2$ likely arose from hydrogen-induced surface roughening due to the increased homologous temperature[28]. Nevertheless, Pd SAED patterns shown in Fig. 3 were the same across different regions within each cross-section and exhibited no discernible changes with annealing conditions, suggesting that Si was not incorporated into the Pd films. The EDS line scan results shown in Fig. S6 also indicate that no Si incorporation occurred in the Pd patterns. This supports the idea that the aforementioned effects of the annealing ambient conditions on facet formation originated from the surface energy anisotropy of Pd.

## Multiscale KMC simulations

The numerical calculation schemes used in this study were designed to capture the aforementioned effects of surface energy anisotropy and

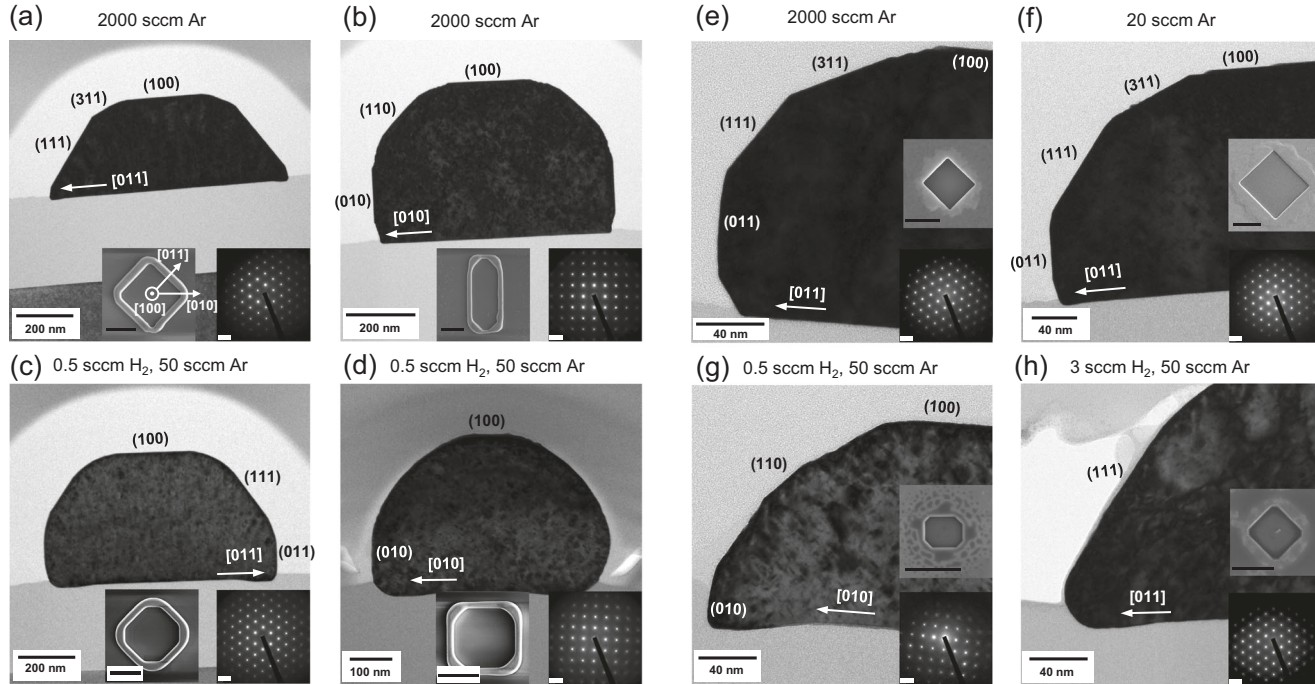

**Fig. 3 | Effects of annealing ambient conditions on the facet morphologies along the edges of ring patterns and holes shown in Fig. 2. a–h** Cross-sectional TEM images of **a–d** ring patterns and **e–h** holes. Insets show top-view SEM images and cross-sectional SAED patterns. Annealing ambient conditions are indicated above each TEM image. The crystallographic directions applicable to all inset SEM images are shown in the inset of (**a**). The facet orientations are indicated in each TEM image. All facet orientations were determined according to the θ−2θ XRD measurement results shown in Fig. 1c and the angles between each facet and the top facets. The SAED patterns confirm the preservation of the single-crystalline nature during the dewetting and the normal directions of the cross sections. Scale bars in the SEM images and SAED patterns of insets represent 3 µm and 6 1/nm, respectively.

gas adsorption on dewetting morphologies by combining the KMC, MD, and DFT methods (see Fig. 4a). Figure 4b, c shows the KMC simulation results corresponding to the dewetting patterns in Fig. 2a, b. The KMC simulation results in Fig. 4b were obtained by setting the values of $\zeta$, $\frac{k_B T}{J(1+4\zeta)}$, and $\frac{E_S}{J(1+4\zeta)}$ $(= E_S^*)$ based on the results of MD simulations for clean Pd, amorphous $SiO_2$, and Pd/$SiO_2$ slabs described in Figs. S7 and S8, where $J$ represents the nearest-neighbor (NN) bond energy, $k_B T$ represents the thermal energy, $E_S$ is the energy difference between film-film and film-substrate NN bonds, and $\zeta$ is the ratio of the next NN (NNN) bond energy to the NN bond energy[15,29]. The NN bond energy $J$ was obtained by calculating the formation energy of two kink pairs using the conjugate gradient minimization method (see Note S1 and Fig. S7 in the Supplementary Information). The canonical ensemble MD simulations provided the values of the surface energies of film ($E_{FV(surface\ orientation)}$) and the film–substrate adhesion energy ($E_A$) at a given temperature, which were used for evaluating $\zeta$ and $E_S^*$ (see the "Methods" section and Figs. S7–S9 for details regarding the MD simulation conditions and procedures). The NVT MD simulation results are presented in Fig. 4d. The values of $\zeta$ and $E_S^*$ were calculated to be 2.422 and 0.85, respectively. The initial film thicknesses were set to three grid units, and the lateral pattern sizes were set such that the lateral size-to-thickness ratios were the same as the experimental values. The temperature $T$ was first multiplied by a correction factor determined using the calculation results of $\zeta$-dependent roughening temperatures, which are based on the statistical mechanics model for step formation. Subsequently, an additional fitting factor $c$ was applied to achieve simulation morphologies that are consistent with experimental results (see the "Methods" section and Note S1 in the Supplementary Information for details). These input parameters determine the hopping rates of the surface atoms in the KMC simulation, as explained in the "Methods" section.

As shown in Fig. 4b, the thin-film patches aligned along the <011> or <001> directions dewet to form uniform ring patterns in the simulation results with $\zeta$ of 2.422. These results are consistent with the experimental results obtained in 0.5 sccm $H_2$ and 50 sccm Ar. As mentioned previously, uniform ring patterns were observed only in the <011> directions when the films were annealed under 2000 sccm Ar. This was associated with the formation of wider {111} facets along the surfaces of the lines aligned along the <011> directions. The formation of wider {111} facets indicates lower energies of the {111} surfaces. In the KMC algorithm, the change in the surface energy anisotropy can be implemented by increasing $\zeta$. The KMC simulation results became more consistent with the experimental results obtained in 2000 sccm Ar as the value of $\zeta$ was increased to 7.5, as shown in Fig. 4c. The increased value corresponds to a surface energy ratio of 1.119 between (100) and (111) surfaces, which is approximately 6% greater than the value at $\zeta = 2.422$. The simulation temperatures were increased and decreased from the values calculated using the statistical mechanics model by multiplying fitting factors greater and smaller than unity for $\zeta = 2.422$ and $\zeta = 7.5$, respectively, to achieve consistent agreement with experimental morphologies. Simulation results without the fitting factor are presented in Fig. S10 of the Supplementary Information. Morphologies obtained without applying the fitting factor were also qualitatively consistent with experimental results regarding the effects of orientation and $\zeta$ on the line stability. However, key morphological characteristics, including the shape and connectivity of the rings and the number of particles, were not accurately reproduced. The morphologies of holes and thickened rims shown in Figs. 2c, d and 3e, g were also more accurately reproduced in the simulations when the values of $\zeta$ and $c$ were set as mentioned above (see Fig. S11 in the Supplementary Information).

The two different values of $\zeta$ are associated with annealing in 0.5 sccm $H_2$/50 sccm Ar and in 2000 sccm Ar, which can lead to

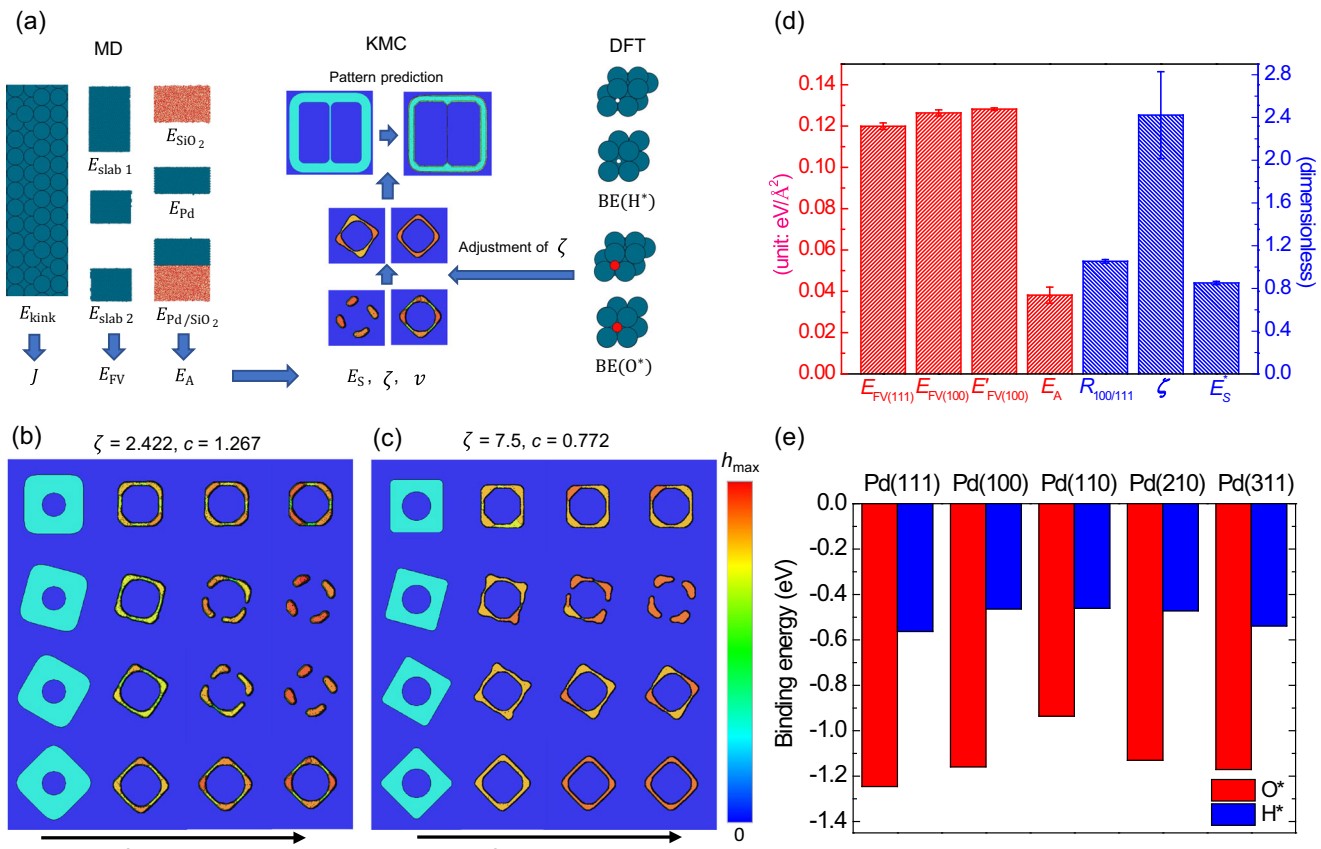

**Fig. 4 | Calculation and simulation results for the dewetting of Pd(100) patches on SiO₂ substrates. a** Schematic illustrating the multiscale calculation method used in this study. **b, c** KMC simulation results for the dewetting of patches shown in Fig. 2a, b. The KMC simulation times in (**b, c**) were different due to the application of the fitting factor. The color bar represents the height in the unit of the simulation grid, with varying maximum values ($h_{max}$) across different columns. For the first column in both (**b, c**), the maximum height is 11. For columns 2–4 in (**b**), the maximum heights are 16, 18, 18, and 16 for rows 1–4, respectively, whereas those in (**c**) are 11 for all rows. The in-plane crystallographic orientations of the patches are the same as those shown in Fig. 2. **d** NVT MD simulation results for setting the

values of $\zeta$ and $E_S^*$. $E_{FV(111)}$ and $E_{FV(100)}$ represent the (111) and (100) surface energies calculated using the slabs shown in Fig. S8a. $E'_{FV(100)}$ and $E_A$ are the (100) surface energy and the adhesion energy calculated using the slabs shown in Fig. S8b, together with the corresponding bulk Pd counterpart. The $R_{100/111}$ represents the ratio of $E_{FV(100)}$ to $E_{FV(111)}$. Error bars represent the standard deviation ($n = 10$). **e** DFT-calculated BEs of O* and H* on different Pd surfaces. All BEs were independently calculated in this study using the same computational setup as in ref. 39, which reported results only for H adsorption on the (111), (100), and (110) surfaces. Source data are provided as a Source Data file.

hydrogen and oxygen incorporation, respectively. The value of $c$ was set to be smaller than unity in the case of $\zeta = 7.5$, indicating that the reduced temperature was higher in the simulations with no fitting factor. This likely arises because the experimental temperature was reduced by the roughening temperature at $\zeta = 0.266$ where the effect of oxygen adsorption was not considered. Oxygen adsorption could influence the roughening temperature of Pd(100), as suggested by the adsorption energetics examined in the following section. A more accurate estimation of roughening temperatures for oxygen-adsorbed Pd(100) surfaces may bring the value of $c$ closer to unity.

It has been shown that a small amount of hydrogen absorption significantly increases the vacancy concentration[30], which implies reduced cohesive energy that would lower the roughening temperature. Compressive stress exerted on the Pd films can also lower the roughening temperature[31] while its relaxation is not a primary driving force for the dewetting process (see Note S2 in the Supplementary Information). Including these effects in the roughening temperature estimation might decrease the value of $c$. The explicit consideration of these effects in estimating the roughening temperature will be investigated in future studies.

It is also noteworthy that key differences between the simulation results in Fig. 4b, c did not originate from the different initial patch shapes. In the simulations, the greater ring stabilities in the directions

adjacent to the <011> directions than those adjacent to the <001> directions were only observed in Fig. 4c, where uniform rings formed only when initial patches were aligned along the <011> directions. These trends remain unchanged when the initial patches in Fig. 4b, c were swapped. (see Figs. S12a, b in the Supplementary Information). The results in Fig. S12c closely reproduced the experimental results in the left panel of Fig. S2a, indicating that the particle formation observed in all directions can also be predicted and explained by the same underlying mechanisms driving the pattern formation in Fig. 2a, b.

### DFT-calculated adsorption energetics and their implications for dewetting

The results shown in Figs. 2 and 3 suggest that the extent of oxidation was greater on the film surfaces along which wider {111} facets were observed. This indicates that oxygen chemisorption could reduce the relative surface energies in the {111} orientations. Gas adsorption changes surface energy anisotropy because the adsorbate binding energy (BE) depends on the crystallographic orientation of the surface. Stronger binding on one surface leads to a greater decrease in the surface energy of the corresponding surface. To investigate whether this effect of gas adsorption is associated with the dewetting results shown in Figs. 2 and 3, we calculated the BEs of chemisorbed O and H

adsorbates (O* and H*) on Pd slabs using spin-polarized DFT calculations and analyzed their dependence on surface orientations (details regarding the DFT calculations are presented in the "Methods" section). As shown in Fig. 4e, the BEs of O* and H* were calculated to be stronger on (111) than on others. They were stronger on (311), which is an intermediate inclined plane along the <011> edges, than on (110) and (210), which are analogous inclined planes along the <001> edges. This stronger binding on (111) and (311) lowers the relative energies of the surfaces compared with those of the other surfaces. The resulting surface energy anisotropy corresponds to a greater value of $\zeta$ in the KMC model, which renders holes bound by <011> edges and leads to the formation of uniform ring patterns only in the <011> directions, as shown in Fig. 4c.

As mentioned in the "Methods" section, the effects of adsorption on the surface energies were characterized by adding a correction term, given by the product of the BEs and the number of adsorbates per unit area, to the surface energies obtained from MD simulations. O and H surface coverages at 800 °C were estimated using Langmuir adsorption theory and assuming dissociative chemisorption (see "Methods" section for details). The surface-energy ratio between the (100) and (111) surfaces was calculated to be -1.119, which corresponds to $\zeta = 7.5$, as the O coverages were -0.0227 and -0.0561 ML on (100) and (111) surfaces at an oxygen partial pressure of -1.10 × 10$^{-3}$ atm at 800 °C, respectively (see Note S3; Figs. S13 and S14 in the Supplementary Information for details).

As shown in Fig. 3a, c, e, f, the (011) facet abuts the (111) facet during the retraction of the <011> edge, indicating that the relative widths of the {111} facets increase as the energy ratio of the (111) surface to the (110) surface decreases. This trend is consistent with the DFT-calculated O* BEs on the (111) and (110) surfaces. Such orientation-dependent binding is likely to cause oxygen chemisorption to significantly decrease the surface energy in the {111} orientations, leading to the formation of wide {111} facets and lagging edge retraction in the <011> directions.

Using an upper-bound estimate based on an Ar–H$_2$ ambient, increasing the H$_2$ flow rate to 3 and 5 sccm in 50 sccm Ar corresponds to partial pressures of -0.0566 and -0.0909 atm, which yield $\zeta$ values of -2.668 and -2.798, respectively. These significantly smaller increases in $\zeta$ compared to the O adsorption case are due to weaker BEs of H* and the resulting lower coverages. This suggests that the effect of hydrogen on the reemergence of the dominant <011> in-plane facets cannot be explained solely by H adsorption, although the less pronounced <011> facets observed under H$_2$-containing ambient conditions are qualitatively consistent with the calculation results (see Fig. S15).

As shown in Fig. 4e, the BE of O* is stronger on Pd(100) than on Pd(210) and Pd(110), meaning that the roughening temperature for Pd(100) could increase by oxygen adsorption. Such stabilization of Pd(100) also provides a plausible mechanism for the upward shift of the roughening temperature considered in the preceding discussion of the fitting factor $c$.

The BEs of S* and C* were calculated to be the weakest on (111) surfaces, as shown in Fig. S16. This further supports that the annealing-ambient dependent hole shapes are not associated with S or C adsorbates.

## Quantitative comparison between experiments and KMC simulations

Figure 5 shows the experimental and simulation results for the dewetting of cross patches with internal circular holes. Annealing was performed in 2000 sccm Ar at 800 °C. The value of $\zeta$ in the KMC simulations was set to 7.5, which led to simulation results that were the most consistent with experimental results for the dewetting of square patches, as shown in Figs. 2b and 4c. As shown in Fig. 5, these initial patches, having internal holes at specific positions and including both concave and convex corners, exhibit dewetting results with stronger dependence on their initial crystallographic alignments compared to those in Fig. 2a, b. These results were also reproduced well by the KMC simulations. For example, the differences in the shape and width of the central ring and arms of the cross patterns between the <001> and <011> directions were consistently observed in the experimental and simulation results (see Fig. S17 in the Supplementary Information for more results).

Figure 6 shows the results of quantitative analyses evaluating the consistency between experimental and simulation results. As shown in Fig. 6a, all feature sizes in the simulated cross pattern shown in Fig. 5b deviated by less than 10% from the corresponding feature sizes measured in the experimental pattern. Most features in the other simulated cross patterns also showed deviations below 10%, although some exceptions were observed. For example, in the cross patterns shown in Fig. 5a, the widths of the arms and body lines were simulated to be significantly larger and smaller, respectively, than those in the experiment. This suggests that the anisotropic edge retraction and hole growth may not be quantitatively modeled solely by the surface energy anisotropy.

Figure 6b presents an analysis of ring patterns with different initial circular hole sizes (see Figs. 2b, 4c, and S18 for the complete sets of patterns). The numbers above each dewetting pattern indicate the rectangularities ($R$) in the interior region of the rings (see Fig. S19 for the details on the measurement). The rectangularities were significantly closer to unity for rings aligned in the <011> directions in both experimental and simulation patterns. The largest deviation was −6.14%, indicating a high quantitative consistency. The ring sizes and widths also deviated by less than 10% in all cases; the largest deviation was 8.33%, observed in the sizes of rings with the smallest initial hole and <011>-aligned external edges.

In the simulation results showing the largest deviations, which were observed in the cross pattern shown in Fig. 5a, the edge retraction in the <001> directions was simulated to be significantly slower than in experiments. This led to noticeably wider arms in the simulation pattern. One possible approach to improving this quantitative deviation would be to include the effects of gas adsorption on the dynamic anisotropy. Figure 6c shows DFT calculation results for the vacancy formation energies on pristine (100) surfaces and those with an O adsorbate (see Fig. S20 in the Supplementary Information for the corresponding results for H, S, and C adsorbates). The vacancy formation energies were significantly lowered by oxygen adsorption. This lowering can render the exchange surface self-diffusion mechanism more favorable because the transition state in the exchange mechanism involves a vacancy beneath a dimer. The formation of a vacancy is not required in the hopping mechanism, which is known to dominate on the (100) surfaces of 3$d$ transition metals[32]. The enhanced exchange diffusion mechanism can therefore lead to faster edge retraction in the <001> directions relative to the <011> directions. Incorporating adsorbate-induced changes in diffusivity on various facets, including the (100) facets, into the KMC model will be needed to improve quantitative accuracy.

## Formation of electrode patterns via lift-off guided by dewetting patterns

Notably, single-crystal metal patterns presented in this study were created on amorphous SiO$_2$ surfaces by inducing dewetting processes on the transferred films. This clearly demonstrates that the surface energy anisotropy of film materials and gas adsorption, rather than the anisotropy of film–substrate interfacial energy that exists on single-crystal substrate surfaces, play critical roles in determining the geometric characteristics of regular dewetting patterns. Furthermore, their ability to create regular dewetting patterns on arbitrary surfaces extends their applicability. For example, metallic patterns formed on thermally oxidized heavily doped Si substrates can be used to create functional structures that constitute back-gated TFTs.

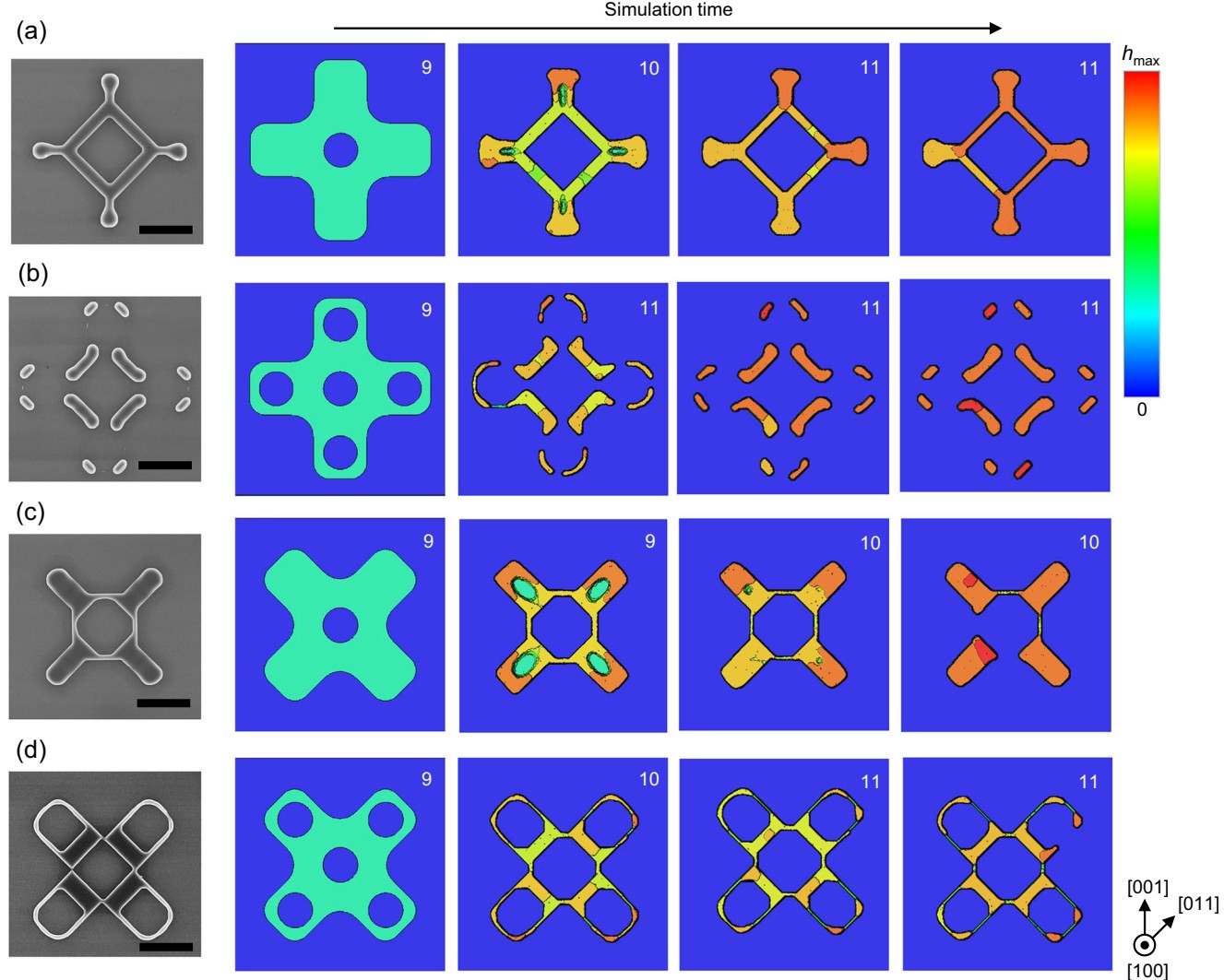

**Fig. 5 | Experimental and simulation results for the dewetting of cross patches patterned from 120-nm-thick Pd(100) films. a–d** SEM images of dewetting patterns and corresponding KMC simulation results. Patches were annealed at 800 °C for 90 min in 2000 sccm Ar. Internal circular holes were initially located in the centers of arms and bodies of the crosses. The KMC simulation times are nearly the same as those in Fig. 4c. The number in the upper-right corner of each result represents the maximum height ($h_{max}$) used for the corresponding color bar. The in-plane crystallographic orientations are indicated in the bottom-right corner. Scale bars in the SEM images represent 5 μm.

We designed initial film patches that could dewet to form line patterns separating the areas of the source, drain, and channel of a TFT; as such, their negative patterns could be used as electrodes. As shown previously, edges remained stable during retraction in the <001> and <011> directions. The retraction was shown to be faster in the <001> directions when the films were annealed in 2000 sccm Ar, which can be beneficial for reducing the line pattern width faster. Hence, the initial patches were designed to align along the <001> directions. Figure 7a shows the KMC simulation results for the designed patches. The simulation was performed using the same input parameters that were employed in the simulations presented in Fig. 5. The central line in the patch evolved to form uniform line patterns whose linewidths decreased over time to the sub-lithographic range. The external edges also retracted without introducing morphological instability. As shown in Fig. 7b, corresponding patterns were experimentally obtained by the dewetting of the Pd patches in 2000 sccm Ar, consistent with the KMC predictions. The sub-lithographic features of the structures were attained by surface self-diffusion from the pattern edge to the top film surface, which increased their heights during

dewetting and made them more suitable as liftoff masks. This process, which is schematically described in Fig. 7c, allows the creation of functional sub-lithographic structures that are unattainable using conventional dewetting processes. Figure 7d shows Au source/drain electrodes with submicron channel lengths, which were formed by liftoff processes involving 50-nm-thick Au film deposition and selective etching of Pd. The formation of channels down to submicron scales demonstrates the controllability of our templated dewetting approach.

To demonstrate the functionality of the patterned electrodes, we fabricated a back-gate IGZO TFT as a proof-of-concept device. Figure 7e presents the output and transfer characteristics of this device. This proof-of-concept demonstration, incorporating a 30-nm-thick IGZO layer and using heavily doped p-type Si substrates with a 300-nm thermal SiO₂ layer as back gates, exhibits clear n-type semiconducting behavior with an ON-OFF ratio of approximately $10^6$ and current saturation[33]. Further optimization of materials and processing conditions is needed to enhance the device performance (additional data is available in Fig. S21 of the Supplementary Information). Nevertheless,

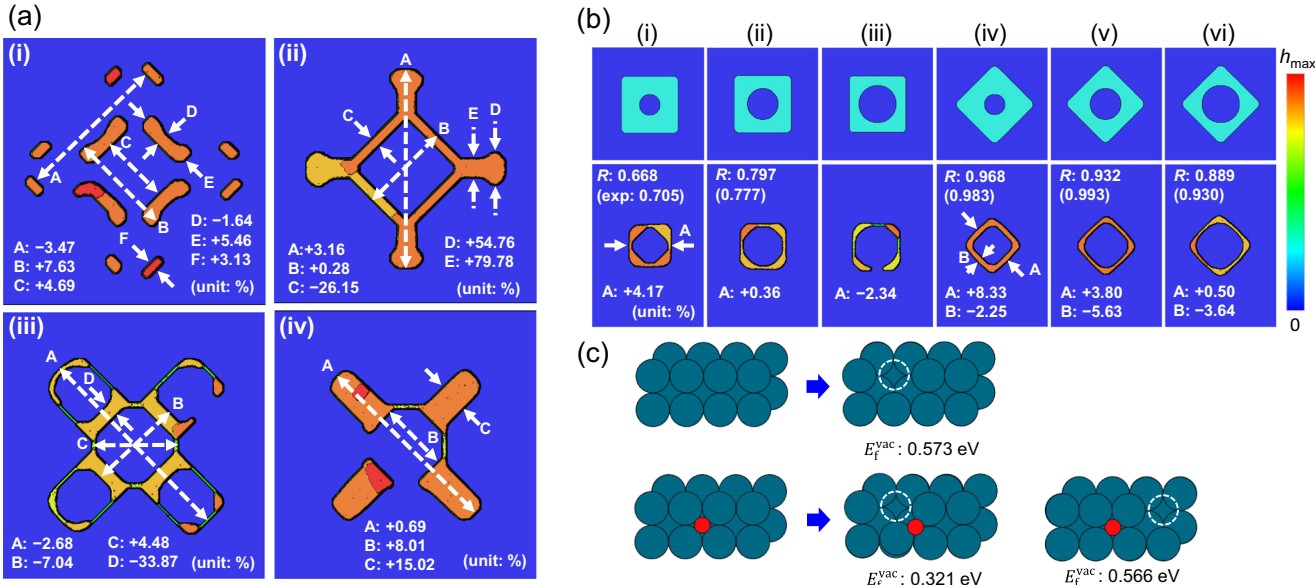

**Fig. 6 | Quantitative analysis of geometric deviations between KMC-simulated and corresponding experimental patterns.** All images shown are simulation results. **a, b** Results for cross and ring patterns, respectively. In (**a**), subpanels (i)–(iv) correspond to the rightmost panels of Fig. 5b, a, d, c, respectively. Feature sizes are indicated with arrows and alphabetical labels. For each pattern, feature-size deviations were calculated as the difference between the simulated and experimental mean feature sizes, normalized by the experimental mean. The values used to determine the mean feature sizes are provided in the Source Data file, and their number varies depending on the number of symmetry-equivalent features and the number of patterns analyzed. In (**b**), each subpanel (i)–(vi) shows an initial pattern (top) and the corresponding KMC-simulated dewetting pattern (bottom) for different initial geometries. $R$ values indicate the rectangularities of the regions inside the ring patterns, calculated by normalizing the inner areas with respect to the enclosing rectangular areas. Values in parentheses represent the experimental results. The KMC parameters used in (**b**) are identical to those in Fig. 4c, and the simulation times are nearly the same as those in the rightmost column of Fig. 4c. The maximum heights ($h_{max}$) used in (**b**) are 11 for all images, except for the dewetting image in (iii), where $h_{max} = 12$. **c** DFT calculation results showing the vacancy formation energies for pristine and oxygen-adsorbed Pd(100) slabs. The dashed circles indicate the vacancy sites. Two distinct vacancy positions were considered for the oxygen-adsorbed slabs. Source data are provided as a Source Data file.

these results demonstrate that the solid-state dewetting processes of pre-patterned single-crystal films can be predictably guided to create structures suitable for the fabrication of functional devices on amorphous substrates.

## Discussion

Improved predictability and applicability could be attained by applying approaches involving multiscale calculations and the transfer of epitaxial films to patterning via the templated solid-state dewetting of single-crystal Pd films. The effects of the surface energy anisotropy and gas adsorption of any material can be theoretically characterized using the proposed calculation scheme and implemented in annealing processes by controlling the ambient conditions. Therefore, the experimental and theoretical approaches used in this study are expected to be applicable to a wide range of film materials.

As mentioned previously, the KMC simulation parameters were determined based on the results of the MD and DFT calculations, through which the effects of the surface energy anisotropy and gas adsorption were incorporated into the dewetting model. This integration, which has not been applied in previous computational studies, is essential for providing process guidelines that are specific to a given film material because these effects exhibit strong material dependence. For instance, Ag(100) and Ag(111) islands grown on Si(100) and Si(111) substrates have been reported to evolve into more isotropic shapes when heated in air than in a hydrogen ambient[34], indicating that oxygen adsorption reduces the surface energy anisotropy. This observation was consistent with DFT-calculated O* binding energies, which were stronger on atomically rough Ag(110) than on Ag(100) and Ag(111), mechanistically consistent with the faceting behavior

discussed for Pd films in this study. Thus, although further refinement is needed to improve quantitative accuracy, the multiscale calculation scheme presented here provides essential insights for designing material-specific dewetting processes to achieve targeted pattern geometries.

In summary, we demonstrated that templated solid-state dewetting processes for single-crystal films can be designed and guided to create specific regular patterns on arbitrary substrates, thereby overcoming the substrate-related limitations reported in previous studies. Single-crystal Pd(100) films, transferred and lithographically patterned on amorphous SiO₂ surfaces, were dewetted to form regular patterns with sub-lithographic features. Their geometric characteristics depended significantly on their initial crystallographic alignments and annealing ambient conditions. Multiscale calculations involving DFT, MD, and KMC methods indicated that the surface energy anisotropy of Pd and gas adsorption play critical roles in determining the geometric characteristics. The surface energy anisotropy of Pd led to the formation of <001> and <011> in-plane facets and changes due to oxygen adsorption, such that the <011> in-plane facets became dominant. The calculation results were supported by the formation of wider (111) facets along the surfaces of dewetting films annealed under ambient conditions, where the extent of oxygen adsorption was greater. Pd patches were designed according to multiscale calculations and dewetted to form line patterns that could be used as liftoff masks to create electrode structures with submicron channels for IGZO TFTs. This study demonstrates the feasibility of predictable and controlled fabrication of functional patterns with regular geometries on arbitrary substrates, enabled by integrating multiscale calculation schemes and film transfer techniques with templated solid-state dewetting of single-crystal films.

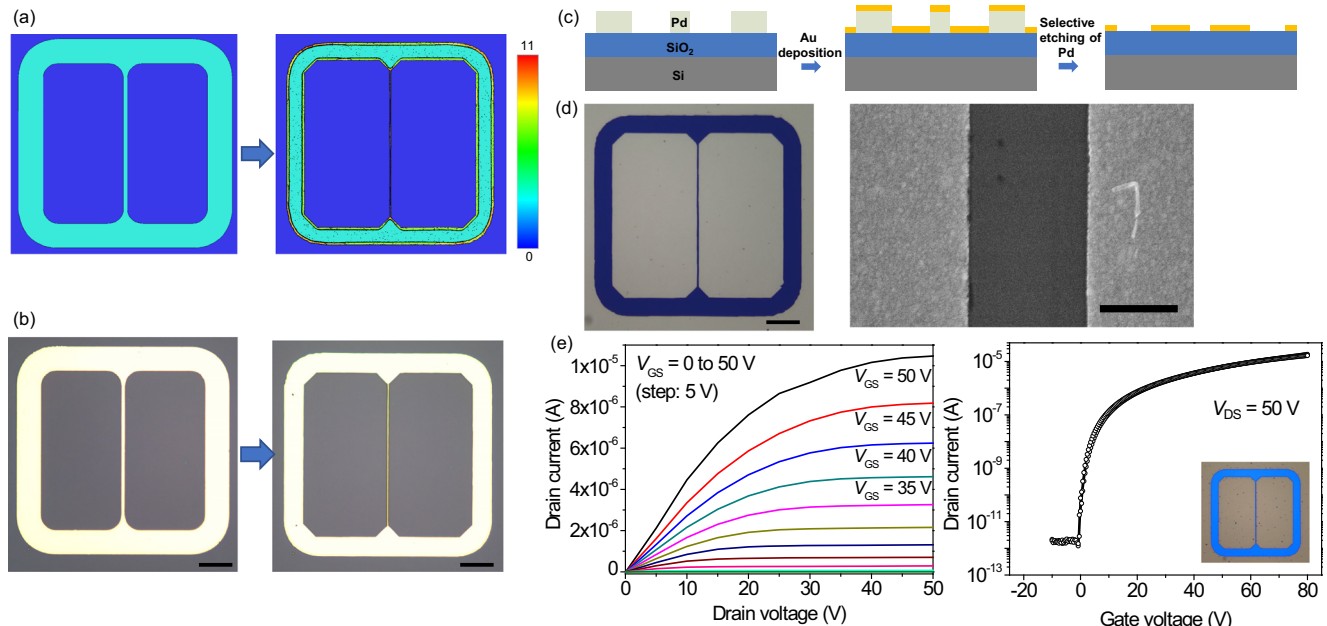

**Fig. 7 | Formation of electrode patterns for thin-film transistors via the dewetting of Pd(100) films and subsequent liftoff of Au. a** KMC simulation results for Pd line patterns with enclosing geometries. The color bar represents the height scale. **b** OM images of Pd line patterns formed by dewetting. Scale bars represent 20 μm. **c** Schematic illustrating the Au liftoff process for creating electrode patterns. **d** (left) OM and (right) SEM images of Au electrode patterns formed by liftoff processes. Scale bars represent 20 μm and 500 nm for the OM and SEM images, respectively. **e** (left) Output and (right) transfer characteristics of an IGZO TFT formed on the same type of Au electrode pattern. The output curves were measured at gate voltages $V_{GS}$ ranging from 0 to 50 V, increasing in 5 V increments from bottom to top. Selected $V_{GS}$ values are labeled beneath the corresponding curves. The transfer curves were measured as a function of gate voltage at a fixed drain–source voltage ($V_{DS}$) of 50 V using a bidirectional gate-voltage sweep. The inset OM image in the transfer curve plot shows a representative device layout of the same type. Source data are provided as a Source Data file.

## Methods

### Thin-film growth and transfer

Epitaxial Pd(100) films with a thickness of 120 nm were deposited on 30-nm-thick Cu(100) buffer layers grown on single-crystal Si(100) substrates via electron-beam (e-beam) evaporation. 99.997% pure Cu and 99.95% pure Pd pellets were used as evaporation sources. The Si substrates were dipped in a 25% HF solution before being introduced into the deposition chamber to strip surface oxides and passivate them via hydrogenation. The chamber was pumped down to a base pressure of high $10^{-7}$ Torr, and the films were deposited at room temperature and at a rate of 0.5 Å/s, which was measured using a quartz crystal sensor. The chamber pressure varied between $1 \times 10^{-6}$ and $4 \times 10^{-6}$ Torr, as measured by an ion gauge.

As mentioned in the main text, the Cu films were deposited not only to enable the epitaxial growth of the Pd films but also to form sacrificial layers selectively etched to separate the epitaxial Pd films for transfer. The as-deposited epitaxial stacks of Pd and Cu films were floated on Cu etchants, which consisted of 2 M ammonium persulfate (reagent grade 98%, Sigma–Aldrich) aqueous solutions, to selectively remove the Cu films. This process resulted in the Pd films floating on the surface of the etchant, while the Si wafer sank. The floating Pd films, after being transferred from the etchant solution to DI water for rinsing, were transferred onto heavily B-doped p-type Si(100) substrates with 300-nm-thick thermal oxide layers. The surfaces of the target substrates were pretreated with $O_2$ plasma to render them hydrophilic, mitigate wrinkling, and facilitate uniform adhesion of the transferred Pd films. The $O_2$ plasma treatment was performed for 30 min in an oxygen plasma cleaner (PDC-32G-2, Harrick Plasma) at a working pressure of $5.8 \times 10^{-1}$ Torr and a radiofrequency power of 18 W.

### Pattern formation via dewetting and liftoff processes

The transferred Pd films were patterned using conventional photolithography, followed by the wet etching of the exposed areas. Wet etching was performed for approximately 1 min in diluted aqua regia solutions comprising 3, 1, and 6 parts of 37% hydrochloric acid, 60%–62% nitric acid, and deionized (DI) water, respectively. The samples were soaked in acetone for 30 min and subsequently rinsed with isopropyl alcohol and DI water.

The patterned and as-transferred Pd films were thermally annealed in a quartz tube furnace at 800 °C for 90 min under various ambient conditions of Ar and $H_2$ to induce solid-state dewetting and create regular patterns. The temperature inside the quartz tube and gas flow rate were measured and controlled using a K-type thermoprobe and mass flow controllers. The flow rates of $H_2$ and Ar were varied to investigate the effects of the extent of gas chemisorption on dewetting.

Fifty-nanometer-thick Au films were deposited on dewetted samples to create negative patterns via liftoff processes. The Au films were also deposited at room temperature via e-beam evaporation under the same conditions employed for the deposition of the Pd and Cu films. The Au films deposited on the Pd patterns were removed by soaking the samples in a wet etchant (Ni etch type 1, Transene Company, Inc.), which selectively etched Pd in approximately 2 min. Short ultrasonication is often needed to enhance the liftoff process.

### Structural and compositional analyses of Pd films and patterns

The crystallinity of the Pd films was characterized using X-ray diffraction (XRD, Phillips X'Pert-PRO) and selective area electron diffraction (SAED) patterns acquired in a transmission electron microscope (TEM, JEOL JEM-2100F). The crystallinity of the dewetting patterns was examined by SAED. Energy-dispersive X-ray spectroscopy (EDS) line-scan data were obtained in the same TEM and analyzed using the Oxford Instruments AZtec software package. The sample preparation for cross-sectional transmission electron microscopy was performed using an FEI Nova NanoLab focused ion beam (FIB). The morphologies and shapes of dewetting patterns and holes were observed using TEM,

scanning electron microscopy (SEM, FEI Nova NanoLab; Hitachi SU8010), and optical microscopy (OM, Olympus BX41M). XPS was performed using an Al Kα X-ray beam (photon energy of 1486.6 eV) in an ultrahigh vacuum spectrometer (Thermo Scientific K-Alpha) to analyze the surface chemical compositions of Pd films annealed under different ambient conditions. Peak deconvolution was performed in CasaXPS using mixed Gaussian–Lorentzian line shapes and a standard Shirley background.

### Formation and electrical characterization of IGZO TFTs

IGZO TFTs were formed by depositing 30-nm-thick IGZO films onto the aforementioned Au electrode patterns via sputtering. The IGZO targets used in the sputtering were formed with an $In_2O_3$:$Ga_2O_3$:ZnO ratio of 1:1:1 (mol%). The films were deposited under a working pressure of $2 \times 10^{-3}$ Torr and a power of 60 W. The sputtering gas atmosphere was maintained at 9 sccm Ar and 1 sccm $O_2$. Prior to electrical measurements, the TFTs were annealed at 350 °C for 1 h in air to remove structural defects in IGZO films without inducing crystallization[33]. Their current–voltage (I–V) characteristics were measured using a probe station (MS TECH, MST5000) and a semiconductor characterization system (Keithley, 4200-SCS) at a pressure of $5 \times 10^{-2}$ Torr.

### DFT calculations

Spin-polarized DFT calculations were performed using the projector augmented-wave method with plane-wave basis sets and the Perdew–Burke–Ernzerhof exchange-correlation functional within the generalized gradient approximation[35] as implemented in GPAW (version 19.8.1 and 22.1.0)[36]. The Atomic Simulation Environment (ASE, version 3.20.1 and 3.22.1) package[37] was used to create model structures for the DFT calculations. The equilibrium lattice constant of Pd was calculated to be 3.946 Å with a plane-wave cutoff energy of 700 eV, an $8 \times 8 \times 8$ Monkhorst–Pack k-point sampling[38]. Slab calculations of Pd(111), Pd(100), and Pd(110) were performed using $2 \times 2 \times 4$ supercells, with the cutoff energy and k-point sampling set to 500 eV and $7 \times 7 \times 1$, respectively, following the parameters used in our previous work[39]. Four-layer slabs containing 16 Pd atoms were used for Pd(311) and Pd(210). Thirty-angstrom-thick vacuum layers were inserted on each side of the slab in the surface normal direction, and a dipole-layer correction was included to prevent interaction between periodic images of the slabs along the surface normal directions. For the Pd bulk and slab calculations, a Fermi–Dirac smearing of 0.1 eV was employed. The self-consistent field convergence criteria were set to $10^{-4}$ eV per valence electron for the energy and $10^{-6}$ electrons per valence electron for the density. The geometries of the molecules, clean slabs, and slabs with adsorbates were optimized using a quasi-Newton algorithm implemented in ASE until the maximum force was ≤0.02 eV/Å. The bottommost layer of the slab was fixed during the geometry optimization. The BEs of the oxygen and hydrogen adsorbates were calculated after geometry optimization as follows:

$$BE\left(O^{*}\right) = E\left(Pd\ slab + O^{*}\right) - E(Pd\ slab) - 0.5 \times E(O_2) \quad (1)$$

$$BE\left(H^{*}\right) = E\left(Pd\ slab + H^{*}\right) - E(Pd\ slab) - 0.5 \times E(H_2) \quad (2)$$

where O* and H* represent the adsorbates, and $E$ represents the total energies of the systems indicated in parentheses.

The surface energies ($E_{surf+O}$) of the Pd planes with O adsorbates were approximated using the following equation:

$$E_{surf+O} = E_{surf} + BE(O^{*}) \times \frac{N_{O^{*}}}{A} \quad (3)$$

where $E_{surf}$ represents the surface energies of the clean Pd planes calculated using MD, and $N_{O^{*}}$ and $A$ represent the number of O

adsorbates and the area of the supercell, respectively. The O surface coverage was calculated as a function of oxygen partial pressure using Langmuir theory as follows:

$$\theta_{O^{\cdot}} = \frac{\sqrt{\exp\left[\left(\mu_{O_2} - 2BE(O^{*})\right)/k_B T\right]}}{1 + \sqrt{\exp\left[\left(\mu_{O_2} - 2BE(O^{*})\right)/k_B T\right]}} \quad (4)$$

where $\mu_{O_2}$ and $BE(O^{*})$ represent the chemical potential of oxygen molecules and the BE of O*, and $k_B$ and $T$ are the Boltzmann constant and temperature. The chemical potentials of gas molecules at 800 °C were estimated using the NIST–JANAF thermochemical data[40], with the tabulated values interpolated and the pressure dependence treated assuming ideal-gas behavior. The calculated surface coverage was used to obtain the value of $\frac{N_{O^{\cdot}}}{A}$. The same calculation was performed for H adsorbates. See Note S3 in the Supplementary Information for details.

DFT calculations for determining the formation energies of vacancy ($E_f^{vac}$) were performed using $4 \times 2 \times 4$ Pd(100) slabs with the same parameters as for the adsorbate binding energies. Clean slabs and those with O, H, S and C adsorbates were used to investigate the effect of adsorption on the vacancy formation energy. The vacancy formation energies were calculated as follows:

$$E_f^{vac} = E(Pd\ slab + vacancy) + 0.25 \times E(bulk) - E(Pd\ slab) \quad (5)$$

where $E(bulk)$ is the total energy of a periodic unit cell consisting of four atoms. The bulk energy was calculated using the same cutoff energy as in the slab calculations and a $12 \times 12 \times 12$ k-point mesh.

### MD simulations

MD simulations were performed using LAMMPS software (version 29 Sep 2021)[41] to obtain the input parameters for the KMC simulations. The model structures for the MD simulations were built using the ASE package. For all ensemble simulations, periodic slabs were used, and their temperatures were equilibrated at 800 °C using Nose–Hoover thermostats. The timestep was always set to 1 fs, and each simulation was run for 100 ps, unless otherwise specified. Vacuum layers of 40–80 Å were inserted at the top and bottom of each slab to model the surfaces. Each ensemble simulation was performed 10 times with different seed numbers to obtain the mean values of the system properties.

The equilibrium lattice constant of Pd at 800 °C and 1 bar was calculated to be 3.943 Å by averaging the results of NPT ensemble simulations in which periodic $20 \times 20 \times 20$ slabs of Pd(100) with no vacuum layer were relaxed for 100 ps. The formation energies of two kink pairs with one- and two-atom heights ($E_{kink1}$ and $E_{kink2}$) were calculated using the conjugate gradient minimization method to obtain the values of $J$ and $\zeta$ in Pd. The structures were relaxed until energy convergence was achieved. As shown in Fig. S7, Pd(1811) slabs with two kink pairs were created by manually moving one or two ledge atoms at a fixed position along a step toward the front of another ledge atom. Formation energies were determined by subtracting the energy of the Pd(1811) slab from that of the Pd(1811) slab with two kink pairs.

To obtain the input parameters for KMC simulations, we ran canonical ensemble simulations of two types of slabs for each of Pd(100) and Pd(111), as shown in Fig. S8a, the total energies of which are denoted as $E_{slab1}$ and $E_{slab2}$, respectively. The surface energies ($E_{FV}$) of Pd were determined from the calculated total energies, as follows:

$$E_{FV} = \frac{1}{2A}\left(E_{slab2} - E_{slab1}\right) \quad (6)$$

where $A$ represents the surface area of one side of the slab.

The total energies of the slabs of Pd(100), amorphous SiO$_2$, and a stack of two ($E_{Pd}$, $E_{SiO_2}$, and $E_{Pd/SiO_2}$) were calculated to obtain the Pd/SiO$_2$ adhesion energies by running canonical ensemble simulations (see Fig. S8b for the structures used in the simulations). The amorphous structure of the SiO$_2$ slab was obtained by running a two-step canonical ensemble simulation, in which a beta quartz slab was heated at 5000 K for 100 ps and subsequently cooled to 300 K in 200 ps. The values of $E_A$ were calculated using the total energy values of the three slabs, as follows:

$$E_A = \frac{1}{A}(E_{Pd} + E_{SiO_2} - E_{Pd/SiO_2}) \tag{7}$$

where $A$ represents the interfacial area. For the slab energy calculations used in Eqs. (6) and (7), the total energies were averaged over the final 10 ps of each simulation.

In all MD simulations, the Pd–Pd interactions were described using a 12-6 Lennard–Jones (LJ) potential with the coefficients reported by Heinz et al.[42] The Si–O and O–O interactions were described by overlaying the Beest–Kramer–van Santen (BKS) potential[43] with an 18-6 LJ potential and the coefficients reported by Yu et al.[44] The cutoffs for the Buckingham and Coulomb interactions in the BKS potential were set to 5.5 and 10 Å, respectively. The long-range part of the Coulomb interaction was computed using the particle–particle particle–mesh method with a relative force accuracy of $1 \times 10^{-4}$. Pd–Si and Pd–O interactions were described using 12-6 LJ potentials. The $\varepsilon$ and $\sigma$ parameters were derived using the geometric-mean mixing rule from the LJ parameters reported by Heinz et al.[42] for Pd–Pd interactions and by Rappe et al.[45] for Si–Si and O–O interactions. The cutoffs were set to $3\sigma$, corresponding to 9.3 Å for Pd–Si and 8.4 Å for Pd–O.

### KMC simulations
The hopping frequency is given by:

$$\nu = \nu_0 \exp\left(\frac{-(n + m\zeta)J + \delta_z E_S}{k_B T}\right) \tag{8}$$

where $n$ and $m$ are the numbers of the nearest and next-nearest neighbors (NN and NNN, respectively), $\zeta$ is the ratio of NNN bond energy to NN bond energy, $J$ represents the NN bond energy, $k_B T$ represents the thermal energy, $E_S$ is the energy difference between film-film and film-substrate NN bonds, and $\delta_z$ is the Kronecker delta. $\delta_z$ is 1 if $z = 0$; otherwise, it is 0. $E_S \approx -a^2 S$ where $S$ and $a$ are the spreading coefficient and the lattice parameter, respectively. The values of $J$, $\zeta$, and $E_S$ were calculated using the following equations:

$$E_{FV(100)}/E_{FV(111)} = (1 + 4\zeta)/\{\sqrt{3}(1 + 2\zeta)\} \tag{9}$$

$$E_S/\{J(1 + 4\zeta)\} = (2E_{FV(100)} - E_A)/2E_{FV(100)} \tag{10}$$

$$J = E_{kink1}/2 \tag{11}$$

where $E_{FV(100)}$ and $E_{FV(111)}$ represent the energies of the (100) and (111) surfaces, respectively[15]. The kink, surface, and adhesion energies in the expressions were calculated, as described in the MD simulation section. The roughening temperature of Pd(100) was estimated by calculating the temperature at which the free energy of an atomic step vanishes in a statistical mechanics model for step formation in a simple cubic lattice. In the model, the step free energy $\beta$ is given as

$$\beta = \frac{1}{2a}\left[J(1 + 2\zeta) - 2k_B T \ln\left[1 + \exp\left(\frac{J(2\zeta - 1)}{4k_B T}\right)\bigg/\sinh\left(\frac{J(1 + 2\zeta)}{4k_B T}\right)\right]\right] \tag{12}$$

where $a$ is the lattice constant, $k_B$ is Boltzmann constant, and $T$ represents the temperature[15,46]. The simulation temperature $T_{KMC}$ was set as follows:

$$T_{KMC} = cT_{exp}\left(\frac{T_r(\zeta)}{T_r(\zeta = 0.266)}\right) \tag{13}$$

where $T_r$ represents calculated roughening temperatures and $T_{exp}$ represents the experimental annealing temperature, which was 1073.15 K. $T_r$ is a function of $\zeta$, and the value of $\zeta$ entered in the denominator was determined to be 0.266 by calculating $E_{kink1}$ and $E_{kink2}$, as mentioned in the MD simulation section. A fitting factor $c$ was applied to obtain KMC simulation results that were further consistent with the experimental results.

### Reporting summary
Further information on research design is available in the Nature Portfolio Reporting Summary linked to this article.

## Data availability
Source data are provided with this paper. The atomic coordinates of the optimized computational models and the initial and final configurations of the MD simulations are available via Zenodo under https://doi.org/10.5281/zenodo.19161526. Source data are provided with this paper.

## Code availability
The custom KMC simulation code used in this study is released under the MIT License and is provided in the software package submitted with this manuscript. It is also available from the corresponding author upon request.

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

## Acknowledgements

We gratefully acknowledge the support from the Basic Science Research Programs (2021R1F1A1061643 (J.Y.) and RS-2024-00353752 (J.Y.)) and the International Research & Development Program (2019K1A3A1A21031220 (J.Y.)) of the National Research Foundation of Korea (NRF) funded by the Ministry of Science and ICT and KISTI (KSC-2023-CRE-0161 (J.Y.)).

## Author contributions

S.J. and S.L. performed the experiments, conducted data analysis, prepared figures, carried out simulations, and contributed to writing and revising the manuscript. D.K., H.K., and J.L. (first) contributed to sample preparation, experimental characterization, and manuscript review. J.L. (second) and W.L. contributed to discussions and manuscript review. O.P. developed the original simulation code and contributed to discussions and manuscript review. J.Y. conceived and supervised the project, acquired funding, performed experiments, calculations, and simulations, contributed to data analysis, prepared figures, and wrote and revised the manuscript.

## Competing interests

The authors declare no competing interests.
