## [Transparent Peer Review File · Nature Communications]

Predictive Patterning via Solid-state Dewetting of Transferred Single-crystal Films

Corresponding Author: Professor Jongpil Ye

Version 0:

Reviewer comments:

Reviewer #1

(Remarks to the Author)

The experiments presented in this paper shed new basic insights into the dewetting of metal films. For example, they show very clearly how the initial orientation of the edges of the films affects dewetting mechanisms. They also demonstrate that edge orientation can dominate the effects of substrate epitaxy. I found the experimental design to be clever and novel. I thus have no hesitation in recommending publication in Nature Communications. I do, however, have some comments that the authors might want to address before publication:

- 1) The annealing that induces dewetting is performed at 800°C. At that temperature I would expect that surface concentrations of relatively weakly adsorbed species such as H (and maybe O) to be very low. The authors could estimate these concentrations from their DFT calculations of adsorption energies. Are the expected concentrations high enough to change the orientation dependent surface energies enough to support their hypothesis about how surface energy affect dewetting?
- 2) Small concentrations of adsorbed species can however have dramatic effects on surface dynamics (by changing defect formation energies) and thus Have the authors considered this possibility? Can they rule it out?
- 3) The presentation of the XPS results were a bit incomplete. What were the surface concentrations of impurities such as C, S and O? Carbon is a very common contaminant in Pd and it would be important to know how much carbon was detected and how it varied with annealing recipe. (And as I mentioned above, trace amounts of O and S can have dramatic effects on surface diffusion.)

(Remarks on code availability)

Reviewer #2

(Remarks to the Author)

This manuscript presents a study on the dewetting of patterned Pd films transferred onto amorphous SiO₂ substrates. Selected configurations are investigated to understand the impact of anisotropies under varying gas-adsorption conditions. A multiscale simulation framework is employed to reproduce experimental observations and provide insights into spontaneous patterning. While the overall process and techniques are well-established in the literature, some elements of novelty may lie in the proposed predictive multiscale framework, which is based on different simulations, and in the approach of transferring Pd films onto amorphous SiO₂, potentially enabling new applications.

The main claim of this work, entering both experiments and simulations, is the achievement of "predictive patterning" via solid-state dewetting. From an experimental standpoint, this claim has been explored in earlier studies (e.g., Ye et al., *Advanced Materials* 23, 1567–1571 (2011)), which demonstrated deterministic outcomes in patterned solid-state dewetting. The simulations based on KMC models effectively incorporate parameters from original MD and DFT calculations, yet they still depend on some experimentally fitted parameters. This partially limits their a priori predictive capability beyond the qualitative level. Quantitative prediction with some fitting parameters basically meets the state of the art of such approaches (e.g., M. Trautmann et al., *Appl. Phys. Lett.* 110, 263105 (2017)) as well as other models. The demonstration related to the self-assembly of the thin-film transistor, as guided by simulation, appears somewhat limited. The simulated dynamics is

relatively short, involve simple morphologies, and do not necessitate the use of KMC simulations (stable dewetting fronts, for example, could be anticipated through independent arguments or dewetting experiments). It is also noted that the formulation of hypotheses concerning the impact of different gas atmospheres appears reasonable but would benefit from clearer organization and presentation. These and other points are discussed in more detail below. Overall, the manuscript and the research presented therein provide some valuable insights into solid-state dewetting and thin film processing. However, I respectfully believe that, at its current stage, a stronger case is needed to justify the central claims for publication in Nature Communications.

More detailed aspects:

- In the introduction, it is noted that one of the key reasons limiting the application of dewetting as a patterning method is the empirical nature of parameters used in KMC simulations, which is claimed to be addressed in this work. I agree that this is a major limitation in predicting the outcome of solid-state dewetting at a level that enables a priori design. Still, the approach presented here relies on an empirical fitting parameter (c) used to rescale T_{KMC} (see the Methods section). The authors transparently report that this is a fitting parameter used to match simulations with experiments, show the results obtained without this correction in the Supplementary Material (which indeed significantly differ from a quantitative point of view), and comment on the need for further research to clarify its origin and role. However, this is in contrast with the initial claim of having resolved the issue of reliance on empirical parameters for quantitative predictions.

- In addition to the previous point, conclusions drawn on reported information show indeed they align with the state of the art and that the main claims are partially supported: i) At line 233, it is stated: "Morphologies obtained without applying the fitting factor were also qualitatively consistent with experimental results ...". Such a qualitative description is already the state of the art in simulating solid-state dewetting and would represent the actual situation before performing dedicated experiments. ii) line 304 - 308: the approach "provide process guidelines for creating patterns ...". This is true and useful. However, obtaining "guidelines" is again what is already known that can be achieved nowadays with numerical simulation in this context. iii) Lines 345-346: "solid-state dewetting processes of pre-patterned single-crystal films can be predictably guided to create structures suitable for functional devices." This is a conclusion reached by other previous experimental studies (as mentioned in the first paragraphs above). The claim here is to introduce a better way based on simulation results, which is only partially true. iv) In the conclusions, "We demonstrated that templated solid-state dewetting processes for single-crystal films can be designed and guided to deterministically create specific regular patterns on arbitrary substrates." This has been demonstrated before, except for the point of the arbitrary substrates. v) End of conclusions: "This study demonstrates the feasibility of the predictable and precisely controllable fabrication of functional regular patterns on arbitrary substrates via the templated solid-state dewetting of single-crystal films." This is for all the aspects mentioned above, only partially supported by the presented evidence.

- The initial shapes of the experimental patterns in Fig. 2(a) and 2(b), which form the basis for the main discussion, appear to differ slightly. While the inset in Fig. 2(a) is well approximated by a square patch with a central circular hole, the inset in Fig. 2(b) shows a patch featuring additional (small) facets at the corners. Why is this the case? Does it have any impact on the results? One would expect that using the same initial shape would be necessary to enable general parametrization and support broader conclusions.

- In Fig. 2, the distinction between rings with regular and irregular shapes is made. While this can be followed qualitatively, quantification would be necessary (as the differences are relatively small anyway). Also, would a longer annealing time also cause these rings to break into islands? Especially the more irregular ones are expected to do so. This makes the discussion in terms of islands vs rings relative to transient phases, which need to be better presented.

- The distinct facets observed at a 0.5 sccm H₂ flow rate in 50 sccm of Ar (Fig. 2d) represent a rather specific case. Could a range of flow rates around this value (e.g., $0.5 \pm ?$) be estimated where this behavior is consistently observed? Additionally, does the stated consistency with multiple other patches refer to patches of identical shape and size, or does it also include variations in geometry?

- The arguments used to explain the experiments (lines 136–181), presented before the multiscale modeling and simulation results, are not very clear and appear too speculative. Phrases such as "the results suggest," "is attributable," "could be," and "consistent with the hypothesis" indicate a lack of clear evidence. This section would benefit from a concise summary of the key points, and rephrasing of critical sentences to clearly distinguish what is directly shown by the experiments and what remains a hypothesis. For instance:

i) Line 144: It is unclear in what sense "These results suggest that oxygen and hydrogen chemisorption favored the formation of $\langle 011 \rangle$ in-plane facets in the Pd(100) films." How are oxygen and hydrogen chemisorption linked to the previously mentioned observations? Should hydrogen chemisorption still be expected under conditions other than 3 sccm?

ii) Lines 146–155: It is not clear how this discussion explains the observations at an H₂ flow rate of 0.3 sccm. In fact, the discussion seems to suggest that $\langle 011 \rangle$ in-plane facets are expected in all cases, which conflicts with the specificity of the observed behavior.

Additional minor comments:

- One aspect that is not discussed is whether the different gas adsorption conditions also influence the dewetting dynamics. In certain cases, dynamical effects characteristic of surface attachment-detachment processes may emerge in addition to

surface diffusion. Could the authors comment on this?

– The interpretation of the results in terms of the surface energy anisotropy of Pd appropriately builds on the framework originally proposed in Ref. 23. Could this be more directly related to the classical Wulff construction/equilibrium shape description?

(Remarks on code availability)

Reviewer #3

(Remarks to the Author)

In this paper, the authors have prepared the single crystalline Pd thin films and transferred on amorphous SiO₂ substrate for the investigation of solid state dewetting behavior. They have also changed the annealing gas conditions with different Ar and H₂ amounts and mixtures. Interesting structures evolutions have been observed. The authors have also tried to trace the dewetting process and behavior via numerical methods. As the authors mentioned, the dewetting of single crystalline metal films have been already studied, and the big difference in this paper is to use amorphous substrate instead of single crystalline MgO substrates. The Results are interesting, but the paper is not clearly formulated, and the novelty is also not high enough for NC. I would suggest to reject and transferred into another journal. Here are some comments:

(1) It is confused in your statement that oxygen and hydrogen chemisorption favored the formation of <011> in-plane facets in the Pd(100) films. Because you have just described that formation of primary <001> in-plane facets only with small amount of H₂.

(2) In figure 3, you said there is no Si in the Pd structure confirmed by SAED. However, it is better to confirm it by investigation with EDX or similar method.

(3) Your study is about the investigation on dewetting of the single crystalline Pd on the amorphous SiO₂ substrate, and suggests that surface energy anisotropy of films and gas adsorption are more important for the structure evolution than the film-substrate interface energy. But how about the influence of strain energy? The thermal extension coefficients between metals and oxides are large.

(4) As you mentioned that the interface energy is not important for the structure evolution due to the amorphous nature, but it is strange that in figure 4b, the structures are clearly different with different tilt angles. Why? Influence of gas flow direction?

(5) The whole arguments about the influence of anisotropy of surface energy and gas adsorption are also not clear enough. By anisotropy of surface energy you mean the orientation dependent aspect. But surface energy is the energy associated with two phases (solid and gas), and this means if you have different gas or gas mixture, then the interface energy is different, and this can also lead to different anisotropy of surface energy.

(Remarks on code availability)

Version 1:

Reviewer comments:

Reviewer #1

(Remarks to the Author)

The authors have thoroughly addressed all of my concerns, and I still recommend publication in Nature Communications. I read the other reviewers concern that the results are not convincing enough for publication in Nature Communications. I think it is true that any computational attempt at predicting the behaviour of surface energies and kinetics is fraught. Surfaces are complex and one can (and should!) always worry that something has been left out (as I stated in my review). But that does not diminish the importance of work such as presented here that makes a serious attempt using all the tools available to account for complex experimental observations. If the presented model of dewetting turns out to still be incomplete, it will be for an interesting reason.

(Remarks on code availability)

Reviewer #2

(Remarks to the Author)

The authors carefully revised their manuscript, addressing most of the comments raised in my previous report, clarifying crucial points, and providing additional evidence. The "stronger case" I was referring to has now been made.

I acknowledge the relevance of making the self-assembly of nanostructures exploiting dewetting compatible with relatively standard processes (room-temperature epitaxial growth, avoiding state-of-the-art vacuum conditions). The proposed

analysis of chemical effects in the gas phase, which is deeper than what has been reported in the literature so far, is also a relevant aspect.

I believe the authors have appropriately downplayed the claim of a fully predictive modeling framework, and I agree that this work represents "a step toward a more predictive modeling framework," although further efforts and understanding (now also discussed in the manuscript) are still needed. It remains, however, that the title points to "predictive patterning of ..." rather than to the aspects mentioned above ("transferred" is relatively unclear at first reading), and that the predictive and reproducible character of dewetting exploiting pre-patterning has already been discussed extensively in the literature (as properly acknowledged by the authors). I therefore suggest making the actual relevant element of novelty more clearly evident from the very beginning (e.g., in the title).

I thank the authors for the extensive explanations concerning KMC simulations and appreciate the effort made to clarify their claims, also through additional discussion in the manuscript. The point raised regarding the proposed "guidelines" is now fully clarified as they indeed refer to the specific process considered here (including ambient conditions) rather than to controlled solid-state dewetting in general. The same applies to remarks on the relevance of the transfer step.

Importantly, quantitative analyses have been added (new Fig. 6). Moreover, the extended simulation campaign, which increases the reliability of the claims, is definitely a plus and demonstrates the robustness of the process, which was not clearly conveyed in the previous draft.

I believe the manuscript can now be accepted for publication, while noting that further tailoring the visibility of the key messages to better convey the actual points of novelty could further increase the work's impact and accessibility.

(Remarks on code availability)

Reviewer #3

(Remarks to the Author)

it is great that the authors have made big efforts to revise the manuscript, and the revisions are properly done. I suggest to accept it for publication. But the language should be polished again. For example, the sentence in the abstract is hard to understand: "The anisotropies of Pd surface energy and gas adsorption strength were found to cause the <001> and <011> in-plane facets to form and favor the latter as oxygen adsorption increases or hydrogen flow rate is sufficiently high."

(Remarks on code availability)

Responses to reviewer comments

We sincerely appreciate the reviewers' insightful comments.

Our detailed, point-by-point responses are provided below.

Revisions of the manuscript made in response to the comments are marked in blue in the revised manuscript.

Author's Note to Reviewers

This note is to inform reviewers that one new figure has been added (now Fig. 6) and several figures have been modified during the revision. The major changes are summarized below.

- Figure 6: A new figure has been added to address the reviewer's request for quantitative analyses regarding the consistency between the experimental results and the KMC simulations.

- All figures including the KMC simulation results: The original images showing the KMC simulation results were saved with gamma correction, which is the default setting in the OpenDX software. For the revised manuscript, all images were saved without gamma correction. Only the color schemes were affected by this change.

- Figure 2: The two OM images in the bottom row of Fig. 2(b) were replaced because the original images were taken from patches with holes slightly smaller than those in the top row. We found this during the quantitative analyses and the images were replaced although the resulting patterns exhibit no appreciable differences. The OM images in Figs. 2(c) and 2(d) were retaken because the original images exhibited a green tint due to a temporary microscope setting at that time. The images have now been replaced to ensure consistent background color and improved clarity. Some panels were additionally updated using newly prepared samples due to additional XPS measurements and hole-shape observations performed during the revision.

The Pd 3d XPS spectra were also remeasured because extending the acquisition range beyond ~347 eV allows more reliable determination of the Shirley background in the presence of plasmon-loss and oxide-satellite features. We therefore extended the measurement range to 355 eV and performed new deconvolutions (the full-range data are not shown in the figure for space limitations but are provided in the source data file). We also added the O 1s spectra (Fig. 2(g)) as suggested by the reviewer.

- Figure 7 (previously Figure 6): The KMC simulation results and OM images in panels (a) and (b) have been replaced due to low resolution. The replacement images were obtained from a different sample fabricated under the same process conditions, and the KMC simulation was reperformed to match its slightly different initial shape. In panel (d), the electrode-pattern images have also been replaced. The previous submission inadvertently included images obtained from 30-nm-thick samples; in the revision, these have been replaced with images from 50-nm-thick samples. 50-nm-thick patterns were used in the device fabrication and characterization as stated in the original manuscript.

- In accordance with the journal's style guidelines, we have made minor adjustments to the figure subpanel labels.

Reviewer #1 (Remarks to the Author):

The experiments presented in this paper shed new basic insights into the dewetting of metal films. For example, they show very clearly how the initial orientation of the edges of the films affects dewetting mechanisms. They also demonstrate that edge orientation can dominate the effects of substrate epitaxy. I found the experimental design to be clever and novel. I thus have no hesitation in recommending publication in Nature Communications. I do, however, have some comments that the authors might want to address before publication:

Response: We appreciate the reviewer's positive and encouraging assessment of our work. We are grateful for the insightful comments, which have helped us substantially improve the clarity and rigor of the manuscript. Our detailed responses to each comment are provided below.

1) The annealing that induces dewetting is performed at 800°C. At that temperature I would expect that surface concentrations of relatively weakly adsorbed species such as H (and maybe O) to be very low. The authors could estimate these concentrations from their DFT calculations of adsorption energies. Are the expected concentrations high enough to change the orientation dependent surface energies enough to support their hypothesis about how surface energy affects dewetting?

Response: We estimated the O and H surface coverages at 800°C based on Langmuir adsorption theory, assuming dissociative chemisorption of O₂ and H₂ molecules. The O coverage was calculated using the following equation:

$$f_o = \frac{\sqrt{\exp(\mu_{O_2} - E_o)/kT}}{1 + \sqrt{\exp(\mu_{O_2} - E_o)/kT}}$$

where μ_{O_2} is the chemical potential of O₂, E_o is the O binding energy, and k and T are the Boltzmann constant and temperature. The same formulation was used for hydrogen. The gas-phase chemical potentials at 800 °C were obtained from the NIST-JANAF thermochemical data via regression analysis, assuming that they are ideal gases. The resulting O and H coverages on Pd(111) and Pd(100) as a function of partial pressure are provided in Fig. S14 in the Supplementary Information.

In the manuscript, we showed that KMC simulation results matched the experimental results in 2000 sccm Ar when ζ was increased to 7.5. This ζ value corresponds to an oxygen partial pressure of $\sim 2.91 \times 10^{-3}$ atm at 800 °C. Under these conditions, the calculated O coverages were ~ 0.056 ML on Pd(111) and ~ 0.023 ML on Pd(100), as now included in the manuscript and Note S2 in the Supplementary Information.

Such low coverages cannot be precisely quantified by ex-situ measurements because native

surface oxides form before and after annealing. Furthermore, the equilibrium partial pressure of oxygen changes with the temperature during the heating and cooling. A definitive quantitative comparison would therefore require fully in-situ dewetting and surface spectroscopy under UHV conditions, which are not currently available to us. Nonetheless, the low O coverages are physically reasonable given the strong purging effect of the high Ar flow rate.

For hydrogen, the calculated coverages at the same partial pressure were substantially lower due to their weaker binding energies, consistent with the reviewer's expectation. Assuming that the tube ambient consists of only argon and hydrogen, increasing the H₂ flow rate to 3 and 5 sccm corresponds to partial pressures of 0.057 and 0.091 atm, which yield ζ values of 2.50 and 2.57, respectively. These increases are significantly smaller compared to the O adsorption case (see Results section: DFT-calculated adsorption energetics and their implications for dewetting).

As shown in Fig. S20(a), hydrogen adsorption lowers the vacancy formation energy on Pd(100), similar to oxygen. This reduction suggests that hydrogen may also influence the dynamic anisotropy by modifying the rates of vacancy-mediated exchange diffusion along the $\langle 001 \rangle$ directions. Nevertheless, the reemergence of the dominant $\langle 011 \rangle$ in-plane facets cannot be explained by H adsorption because the magnitude of this effect is also expected to be modest.

2) Small concentrations of adsorbed species can however have dramatic effects on surface dynamics (by changing defect formation energies) and thus Have the authors considered this possibility? Can they rule it out?

Response: We agree that the effect raised by the reviewer cannot be fully ruled out within the scope of the present study. We addressed this possibility by calculating the vacancy formation energies on the Pd(100) surface using DFT, considering multiple vacancy-adsorbate configurations with $4 \times 2 \times 4$ slabs (see Methods). The configurations examined in the calculations are shown in Fig. 6(c) of the main text and Fig. S20 in the Supplementary Information.

As shown in Figs. 6(c) and S20, both O and H adsorption reduce the vacancy formation energy on Pd(100). For the nearest-neighbor configuration, the vacancy formation energy decreases from 0.573 eV to 0.321 eV for O adsorption and to 0.529 eV for H adsorption, with smaller reductions for more distant configurations (0.566 and 0.565 eV). Because exchange-mediated diffusion along the $\langle 001 \rangle$ directions on fcc(100) proceeds through vacancy formation, these reductions indicate that adsorbates can enhance surface diffusion rates along the $\langle 001 \rangle$ directions. The effect is expected to be stronger for oxygen due to the larger reduction in vacancy formation energy.

These changes in exchange-mediated diffusion suggest that adsorbates may increase the relative surface diffusivity and edge-retraction rates along the $\langle 001 \rangle$ directions compared to the $\langle 011 \rangle$ directions, with oxygen having a stronger effect.

Such effects may contribute to the directional retraction behavior observed experimentally, but a fully quantitative assessment would require incorporating adsorbate-specific diffusion barriers and vacancy energetics on various facets into the KMC model—extensions that are beyond the scope of the present study. Nonetheless, the DFT results indicate that small concentrations of adsorbates can modify defect formation energies, providing a physically grounded basis for their potential influence on surface dynamics.

3) The presentation of the XPS results were a bit incomplete. What were the surface concentrations of impurities such as C, S and O? Carbon is a very common contaminate in Pd and it would be important to know how much carbon was detected and how it varied with annealing recipe. (And as I mentioned above, trace amounts of O and S can have dramatic effects on surface diffusion.)

Response: We performed additional XPS measurements and analyses for samples annealed under the ambient conditions used in Fig. 2. The corresponding O 1s, C 1s, and S 2p spectra are now provided in Fig. 2(g) (O 1s) of the main text and Fig. S4 (C 1s and S 2p) in the Supplementary Information. As shown in Fig. 2(g), the relative intensity of the O 1s peak associated with Pd surface oxides and oxygen adsorbates increases as the Ar flow rate decreases, consistent with the increasing fractions of the corresponding components in the revised Pd 3d spectra in Figs. 2(e) and 2(f).

All C 1s spectra exhibit typical signatures of adventitious carbon (Fig. S4), indicating that the detected carbon originates primarily from ambient contamination. The S 2p spectra were fitted using three doublets corresponding to terminal S in polysulfides, central S in polysulfides, and sulfates. The sulfur atomic concentration generally ranged from 0 to 2 at% in survey scans. Since most sulfur species were identified as polysulfides, the number of S atoms directly bonded to the Pd surface should be smaller than the total atomic percent.

Although these impurity levels are not negligible, we found no evidence that sulfur affected the hole morphology. These contaminants can shift the Pd 3d peak to a higher-binding-energy region, potentially increasing the Pd^{δ+} peak intensity. We mentioned in the manuscript that the Pd²⁺ (PdO) peak is therefore a better indicator of oxidation (see Results section: Effects of Pd surface energy anisotropy and oxygen adsorption).

To further examine whether S or C adsorption could influence surface energy anisotropy or dynamic anisotropy, we performed additional DFT calculations. As shown in Figs. S16(a–c), the binding energies of S and C are the weakest on Pd(111) and strongest on Pd(100), opposite to the trend for O and H. This implies that S and C adsorption would modify the surface energy anisotropy in the opposite direction and would not promote the formation of the prominent (111) facets observed in Fig. 3. Therefore, neither the orientation-dependent hole shape nor the ring stability in Fig. 2 can be explained by S or C adsorption.

Figure S20 shows that, unlike O and H, S and C adsorption increase the vacancy formation energy on Pd(100) (to 0.877 and 0.778 eV for the nearest configurations, with smaller increases of 0.595 and 0.599 eV for farther ones). Such increases would reduce, rather than enhance,

vacancy-mediated diffusion on Pd(100).

Reviewer #2 (Remarks to the Author):

This manuscript presents a study on the dewetting of patterned Pd films transferred onto amorphous SiO₂ substrates. Selected configurations are investigated to understand the impact of anisotropies under varying gas-adsorption conditions. A multiscale simulation framework is employed to reproduce experimental observations and provide insights into spontaneous patterning. While the overall process and techniques are well-established in the literature, some elements of novelty may lie in the proposed predictive multiscale framework, which is based on different simulations, and in the approach of transferring Pd films onto amorphous SiO₂, potentially enabling new applications.

The main claim of this work, entering both experiments and simulations, is the achievement of "predictive patterning" via solid-state dewetting. From an experimental standpoint, this claim has been explored in earlier studies (e.g., Ye et al., *Advanced Materials* 23, 1567–1571 (2011)), which demonstrated deterministic outcomes in patterned solid-state dewetting. The simulations based on KMC models effectively incorporate parameters from original MD and DFT calculations, yet they still depend on some experimentally fitted parameters. This partially limits their a priori predictive capability beyond the qualitative level. Quantitative prediction with some fitting parameters basically meets the state of the art of such approaches (e.g., M. Trautmann et al., *Appl. Phys. Lett.* 110, 263105 (2017)) as well as other models. The demonstration related to the self-assembly of the thin-film transistor, as guided by simulation, appears somewhat limited. The simulated dynamics is relatively short, involve simple morphologies, and do not necessitate the use of KMC simulations (stable dewetting fronts, for example, could be anticipated through independent arguments or dewetting experiments). It is also noted that the formulation of hypotheses concerning the impact of different gas atmospheres appears reasonable but would benefit from clearer organization and presentation. These and other points are discussed in more detail below. Overall, the manuscript and the research presented therein provide some valuable insights into solid-state dewetting and thin film processing. However, I respectfully believe that, at its current stage, a stronger case is needed to justify the central claims for publication in *Nature Communications*.

We appreciate the reviewer's thoughtful assessment and constructive comments, which have helped us improve the clarity of the manuscript. Before addressing the specific points, we briefly clarify how this work extends previous studies on patterned solid-state dewetting.

As the reviewer mentioned, the formation of deterministic patterns through the solid-state dewetting of single-crystal films has been previously demonstrated (e.g., Ye et al., *Adv. Mater.* 2011), where the role of surface energy anisotropy was effectively established. However, those approaches relied exclusively on single-crystal MgO substrates, which require long preheating

for epitaxial growth and are sensitive to vacuum conditions. These practical constraints have limited the broader use of the method for device-relevant structures.

In this work, we eliminate these constraints by using Si substrates for room-temperature epitaxial growth of Cu and Pd, followed by transfer onto amorphous SiO₂. This removes the need for substrate preheating and makes the process compatible with commonly available e-beam evaporation systems. We believe this substantially improves the accessibility of anisotropy-guided dewetting for functional device fabrication.

From a scientific perspective, a central contribution of this study is the identification of the chemical origin of ambient-dependent morphological changes. To our knowledge, previous solid-state dewetting studies did not incorporate DFT calculations of adsorbate binding energies. These calculations enable us to quantify how gas adsorption modifies the surface energy anisotropy, and they provide the necessary energetic input for our multiscale simulation framework.

Our KMC simulations incorporate MD simulations and DFT calculations, allowing for variations in the surface energy anisotropy. Earlier KMC frameworks did not explicitly include such anisotropy-modifying effects, which limited their applicability across different film materials and processing conditions. While we acknowledge that the fitting parameter c prevents fully quantitative predictions, the integration of DFT-based energetic trends represents a step toward a more predictive modeling framework.

Finally, the use of amorphous substrates allows us to exclude interfacial anisotropy effects and isolate the contributions from surface energy anisotropy and gas adsorption. We agree with the reviewer that some aspects of the presentation required clarification, and we have revised the manuscript accordingly in response to the detailed comments listed below.

More detailed aspects:

- In the introduction, it is noted that one of the key reasons limiting the application of dewetting as a patterning method is the empirical nature of parameters used in KMC simulations, which is claimed to be addressed in this work. I agree that this is a major limitation in predicting the outcome of solid-state dewetting at a level that enables a priori design. Still, the approach presented here relies on an empirical fitting parameter (c) used to rescale T_{KMC} (see the Methods section). The authors transparently report that this is a fitting parameter used to match simulations with experiments, show the results obtained without this correction in the Supplementary Material (which indeed significantly differ from a quantitative point of view), and comment on the need for further research to clarify its origin and role. However, this is in contrast with the initial claim of having resolved the issue of reliance on empirical parameters for quantitative predictions.

Response: We agree with the reviewer that our approach does not fully enable quantitative a priori prediction because the fitting parameter c is still required, as clearly noted in the manuscript. Although we could not eliminate this parameter, we discussed its possible origins in the main text and have revised our claims in the introduction to avoid implying that this issue has been completely resolved.

We think that the need for c primarily arises from the fact that adsorption and absorption effects were not included when determining the roughening temperature used to normalize the experimental temperature. In Fig. 4(c), the fitted value of $c = 0.772$ indicates that T_{KMC} was overestimated. This is likely because the experimental temperature (800 °C) was normalized using the roughening temperature at $\zeta=0.266$, which was obtained from total-energy calculations of Pd(1811) slabs containing a single-kink pair of one- and two-atom height. In these calculations, the effect of adsorption was not considered. As shown in Fig. 4(e), the BE of oxygen was calculated to be stronger on (100) than on (210) and (110) surfaces, suggesting that oxygen adsorption could increase the roughening temperature of (100) surfaces. Taking this effect into account when normalizing the experimental temperature could make the value of c closer to unity.

In contrast, the fitted value of $c = 1.267$ in the left panel of Fig. 3(a) indicates that T_{KMC} was underestimated. Absorbed hydrogen is known to significantly increase the vacancy concentration, implying a reduced cohesive energy and, consequently, a lower roughening temperature. It is also known that compressive stress lowers the roughening temperature. Considering these effects in the estimation of the roughening temperature would again tend to move the fitted value of c closer to unity.

These potential origins of the fitting parameter are discussed in both the main text and the Supplementary Information of the revised manuscript. Incorporating adsorption- and absorption-dependent effects, as well as stress effects, into the multiscale framework would require developing new force fields for MD simulations and extending the KMC algorithm, which is substantial future work. To reflect these limitations more accurately, we have revised the introduction to state that this work improves the limitations of previous simulation frameworks rather than fully resolving the reliance on empirical parameters (see Results section: Multiscale KMC simulations and DFT-calculated adsorption energetics and their implications for dewetting).

- In addition to the previous point, conclusions drawn on reported information show indeed they align with the state of the art and that the main claims are partially supported: i) At line 233, it is stated: "Morphologies obtained without applying the fitting factor were also qualitatively consistent with experimental results ...". Such a qualitative description is already the state of the art in simulating solid-state dewetting and would represent the actual situation before performing dedicated experiments. ii) line 304 - 308: the approach "provide process guidelines for creating patterns ...". This is true and useful. However, obtaining "guidelines" is again what is already known that can be achieved nowadays with numerical simulation in this context. iii) Lines 345-346: "solid-state dewetting processes of pre-patterned single-crystal films can be predictably guided to create structures suitable for functional devices." This is a conclusion reached by other previous experimental studies (as mentioned in the first paragraphs above). The claim here is to introduce a better way based on simulation results, which is only partially true. iv) In the conclusions, "We demonstrated that templated solid-state dewetting processes for single-crystal films can be designed and guided to deterministically create specific regular patterns on arbitrary substrates." This has been demonstrated before, except for the point of the arbitrary substrates. v) End of conclusions: "This study demonstrates the

feasibility of the predictable and precisely controllable fabrication of functional regular patterns on arbitrary substrates via the templated solid-state dewetting of single-crystal films." This is for all the aspects mentioned above, only partially supported by the presented evidence.

Response: We respond to each point below.

i) We agree that qualitative trends obtained without the fitting parameter can already be reproduced using the state of the art. However, in previous approaches—including the work mentioned by the reviewer (Appl. Phys. Lett. 110, 263105 (2017))—all key parameters such as J and E_s were empirically determined, and no theoretical component accounting for the surface energy anisotropy was implemented. This limits their applicability to different film materials and processing conditions.

A recent work has introduced ζ to reproduce a wide range of experimental hole morphologies. Our work further shows that the effects of gas adsorption on surface energy anisotropy can be incorporated by using ζ within a multiscale calculation framework, and that ζ can be linked to adsorbate properties using DFT. In the above-mentioned recent theoretical work (ref. 15), the value of ζ was chosen arbitrarily. In contrast, our multiscale framework incorporates the chemical properties of the film materials and gas species, allowing it to be applied to different materials and annealing conditions.

We agree that the fitting parameter must ultimately be removed for fully quantitative predictions. Instead of removing it, we discussed the potential origins of the parameter c in each case. Accordingly, we have revised our introduction to emphasize the implementation of gas-adsorption effects on the surface energy anisotropy within the multiscale framework, rather than suggesting that all empirical parameters were eliminated.

ii) State-of-the-art numerical approaches cannot provide correct guidelines on the effect of annealing ambient because they do not incorporate material-dependent adsorbate binding energies. The orientation-dependence of binding energies vary strongly with the film material. For instance, O binding energies strengthen in the order of Ag(110), Ag(100) and Ag(111) as previously reported by the corresponding author—opposite to the Pd case shown in this work. As demonstrated in ref. 35, Ag(100) and Ag(111) islands annealed in air became more isotropic, consistent with the relative binding strengths (with O binding being stronger on atomically rough surfaces for Ag) (see Discussion section).

We are currently extending our multiscale approach to other materials such as nickel and gold. For Ni films on MgO, previous studies suggested that oxygen adsorption increases the relative width of the (110) facet. We think this may also be explained by stronger O binding energies on (110) than on (100) and (111).

While more systematic calculations and experiments are needed to generalize this point, we believe that incorporating DFT-calculated adsorbate binding energies into the multiscale framework is essential for providing accurate guidelines. In this sense, we respectfully disagree that our guidelines correspond to what can already be obtained by existing numerical simulations.

iii) We have revised the sentence as suggested by the reviewer to clarify the contribution of this work, specifically the ability to perform predictable patterning on arbitrary substrates:

(original) solid-state dewetting processes of pre-patterned single-crystal films can be predictably guided to create structures suitable for functional devices

(revised) solid-state dewetting processes of pre-patterned single-crystal films can be transferred and predictably guided to create structures suitable for functional devices on arbitrary substrates.

This reflects the fact that previous studies were limited to substrates that enabled epitaxial growth, whereas our transfer-based approach removes this constraint.

iv) We have cited previous works and clarified the contributions of this study. The main contributions are (1) implementing the multiscale calculation scheme and (2) forming single-crystal patterns on amorphous substrates. We revised the sentence accordingly:

(original) We demonstrated that templated solid-state dewetting processes for single-crystal films can be designed and guided to deterministically create specific regular patterns on arbitrary substrates

(revised) We demonstrated that templated solid-state dewetting processes for single-crystal films can be designed and guided to deterministically create specific regular patterns on amorphous substrates, overcoming the substrate limitations of previous studies.

v) We revised the concluding statement to more accurately reflect the contributions of this work:

(original) This study demonstrates the feasibility of the predictable and precisely controllable fabrication of functional regular patterns on arbitrary substrates via the templated solid-state dewetting of single-crystal films

(revised) This study demonstrates the feasibility of predictable and precisely controllable fabrication of functional regular patterns on arbitrary substrates through the integration of multiscale calculation schemes and film transfer techniques with templated solid-state dewetting of single-crystal films

– The initial shapes of the experimental patterns in Fig. 2(a) and 2(b), which form the basis for the main discussion, appear to differ slightly. While the inset in Fig. 2(a) is well approximated by a square patch with a central circular hole, the inset in Fig. 2(b) shows a patch featuring additional (small) facets at the corners. Why is this the case? Does it have any impact on the results? One would expect that using the same initial shape would be necessary to enable general parametrization and support broader conclusions.

Response: The slight differences in the initial shapes of the patches in Figs. 2(a) and 2(b) arose from minor variations in the photoresist exposure, development, or wet-etching steps. Such variations are common at these feature sizes (6–8 μm edge length and 2–5 μm hole diameter). As the reviewer noted, the precise initial geometry can influence the details of the

morphological evolution during dewetting.

Despite these small geometric differences, our main conclusions rely on two key findings that are robust across different initial shapes:

(1) In 2000 sccm Ar, the ring stability is strongly direction-dependent. Uniform rings appear only along the $\langle 011 \rangle$ direction, and patches aligned along $\langle 011 \rangle$ and its adjacent directions show higher stability than those aligned along $\langle 001 \rangle$ and its adjacent directions.

(2) In 0.5 sccm H₂ and 50 sccm Ar, the dewetting also remains direction-dependent, but the stability contrast between the two direction groups is much smaller, and both $\langle 001 \rangle$ - and $\langle 011 \rangle$ -aligned patches are relatively stable under these conditions.

To test whether the initial shape affects these conclusions, we examined patches whose initial geometries were intentionally or unintentionally varied from those in Figs. 2(a) and 2(b). The results (Fig. S2) were consistent with the two key findings above. This indicates that the slight geometric deviations do not alter the fundamental physics underlying the direction-dependent ring stability (see Results section: Ambient-dependent dewetting behaviors of transferred Pd(100) films).

We also repeated the KMC simulations in Fig. 4 after swapping the initial patch geometries (Figs. S12(a) and S12(b)). The simulated results remained consistent with our main conclusions. In Fig. S12(b), only the part of the second key finding related to the relatively higher stability of the $\langle 001 \rangle$ and $\langle 011 \rangle$ directions cannot be observed, because all patches decay into particles under that condition. Nevertheless, the reduction in the stability contrast between the two direction groups—which is the main implication of the second key finding—is still reproduced. This behavior is consistent with the experimental observation that patches annealed in 0.5 sccm H₂ and 50 sccm Ar decay into particles (Fig. S2) and with the corresponding KMC results in Fig. S12(c) (see Results section: Multiscale KMC simulations).

Therefore, both the experimental data and the simulation results confirm that the slight differences in initial patch shape do not compromise the generality of our conclusions. We retained the patches in Fig. 2 because they most clearly highlight the two key findings and provided the additional cases in Fig. S2 for completeness. This clarification is now included in the revised manuscript (see the two Results subsections referenced above).

- In Fig. 2, the distinction between rings with regular and irregular shapes is made. While this can be followed qualitatively, quantification would be necessary (as the differences are relatively small anyway). Also, would a longer annealing time also cause these rings to break into islands? Especially the more irregular ones are expected to do so. This makes the discussion in terms of islands vs rings relative to transient phases, which need to be better presented.

Response: We agree that quantitative analysis is necessary. Figure 6 presents quantitative measurements of feature sizes and geometric characteristics for the patterns in Fig. 2(b), Fig. 4(c), Fig. 5, and Fig. S18. As shown in Fig. 6, the deviations between the simulated and

experimental values are mostly within 10%. These results were obtained using the same simulation conditions as in Figs. 4(c) and 5, including the fitting factor. As noted earlier, achieving comparable quantitative accuracy without the fitting factor will require an improved method for determining the roughening temperature.

Some features exhibit relatively larger deviations. We suggested that incorporating dynamic anisotropy into the KMC algorithm could reduce these discrepancies. To support this, we provided DFT results in Fig. 6(c), showing that oxygen adsorption lowers the vacancy formation energy on Pd(100). Because exchange diffusion on fcc(100) surfaces requires vacancy formation whereas hopping diffusion does not, this reduction indicates a relative enhancement of exchange diffusion along the $\langle 001 \rangle$ directions. Accounting for adsorbate-induced diffusivity changes across multiple facets in the KMC model may improve quantitative accuracy.

Regarding transient versus later-stage morphologies, we added additional experimental data for ring patterns in Fig. S2. Certain aspects of the orientation-dependent evolution can indeed be illustrated more clearly in later-stage patterns—for example, misaligned patches with larger initial holes ultimately break, as shown in Figs. S2. However, the transient-stage patterns in Fig. 2 most clearly reveal the contrast between the two misaligned directions. Under 2000 sccm Ar, the misaligned direction closer to $\langle 011 \rangle$ is significantly more stable, whereas this anisotropic behavior is not observed under 0.5 sccm H₂ and 50 sccm Ar.

These quantitative results and explanations have been incorporated into the revised manuscript (see Results section: Quantitative comparison between experiments and KMC simulations).

- The distinct facets observed at a 0.5 sccm H₂ flow rate in 50 sccm of Ar (Fig. 2d) represent a rather specific case. Could a range of flow rates around this value (e.g., $0.5 \pm ?$) be estimated where this behavior is consistently observed? Additionally, does the stated consistency with multiple other patches refer to patches of identical shape and size, or does it also include variations in geometry?

Response: We performed thermal annealing under various H₂ flow rates with 50 sccm of Ar. As shown in Fig. S3, the $\langle 001 \rangle$ edges were primary when the hydrogen flow rate was 0.5 sccm. Their relative lengths gradually decreased as the flow rate increased from 0.5 to 2 sccm, and became negligible at 3 sccm and above. This is mentioned in the revised manuscript (see Results section: Effects of Pd surface energy anisotropy and oxygen adsorption).

Regarding the consistency across multiple patches, this refers to patches with minor variations in initial geometry (as noted in our earlier response concerning photolithography and wet-etching fluctuations). Despite these small geometric differences, the fundamental relationship between patch orientation and ring stability remained the same.

– The arguments used to explain the experiments (lines 136–181), presented before the multiscale modeling and simulation results, are not very clear and appear too speculative. Phrases such as “the results suggest,” “is attributable,” “could be,” and “consistent with the

hypothesis” indicate a lack of clear evidence. This section would benefit from a concise summary of the key points, and rephrasing of critical sentences to clearly distinguish what is directly shown by the experiments and what remains a hypothesis. For instance:

i) Line 144: It is unclear in what sense “These results suggest that oxygen and hydrogen chemisorption favored the formation of <011> in-plane facets in the Pd(100) films.” How are oxygen and hydrogen chemisorption linked to the previously mentioned observations? Should hydrogen chemisorption still be expected under conditions other than 3 sccm?

ii) Lines 146–155: It is not clear how this discussion explains the observations at an H₂ flow rate of 0.3 sccm. In fact, the discussion seems to suggest that <011> in-plane facets are expected in all cases, which conflicts with the specificity of the observed behavior.

Response: We appreciate the reviewer’s comment regarding the clarity of this section.

i) We agree that the earlier paragraph was unclear because it preceded the discussion where the evidence for adsorbate-induced faceting is presented. We have therefore revised this paragraph to remove statements implying that oxygen and hydrogen chemisorption directly promote specific in-plane facet orientations at this stage of the paper. Instead, we now separate the experimentally supported observations from the hypotheses. The experimental results are presented as they are, and any interpretation involving adsorbate coverage or adsorbate-induced surface energy anisotropy is explicitly stated as a hypothesis that requires further evidence.

The earlier ambiguity also arose from the insufficient explanation of hydrogen’s dual roles:

- (1) acting as a reducing agent that removes surface oxides, and
- (2) acting as an adsorbate that may influence surface energy and surface diffusion anisotropy.

This distinction has now been clarified in the revised manuscript.

ii) We also agree that the explanation of the behavior at 0.5 sccm H₂ required clarification. (We assume the reviewer’s reference to 0.3 sccm was meant to refer to 0.5 sccm.) At such a low flow rate, most of the supplied hydrogen is consumed in reducing residual surface oxides, leaving only a small fraction available for adsorption. As a result, the effect of hydrogen adsorption on the surface energy anisotropy becomes limited, and <001> edges remain primary under these conditions. When the H₂ flow rate is sufficiently increased, oxide reduction becomes complete and excess hydrogen remains available for adsorption, at which point adsorption-induced changes in anisotropy can emerge. This distinction has been clarified in the revised paragraph, with hypotheses clearly separated from experimental evidence.

We also found that the low BE of H leads to a very low surface coverage at 800 °C in the calculation suggested by another reviewer. Our revised XPS analyses also indicate that there is no evidence that hydrogen adsorption plays a critical role at higher H₂ flow rates. Hence, we revised our claim to state that the origin of the reemergence of the <011> facets needs to be further investigated using in-situ surface measurements (see Results section: Effects of Pd surface energy anisotropy and oxygen adsorption and DFT-calculated adsorption energetics

and their implications for dewetting).

Additional minor comments:

- One aspect that is not discussed is whether the different gas adsorption conditions also influence the dewetting dynamics. In certain cases, dynamical effects characteristic of surface attachment-detachment processes may emerge in addition to surface diffusion. Could the authors comment on this?

Response: We agree that gas adsorption can influence the dewetting dynamics. As shown in Fig. 6(c) and Fig. S20, gas adsorption modifies the vacancy formation energy on Pd(100). This affects the rate and possibly the dominant mechanism of surface self-diffusion. Oxygen and hydrogen adsorption decrease the vacancy formation energy, suggesting that exchange self-diffusion could be facilitated, whereas sulfur and carbon adsorption increase it, implying a reduction in the exchange-diffusion rate. We have discussed the potential implications of these adsorbate-induced changes on pattern evolution in the main text and in earlier responses (see Results section: Quantitative comparison between experiments and KMC simulations).

At this stage, we cannot determine how strongly these dynamical effects influence the rim morphology. Although both oxygen and hydrogen lower the vacancy formation energy, a definitive identification of the dominant mass-transport mechanism would require a more detailed comparison of the possible kinetic pathways. In our experiments, the rim morphologies developed in a consistent manner across all ambient conditions, suggesting that surface diffusion remained the primary mechanism under our processing conditions. This clarification has been added to the revised manuscript (see Results section: Effects of Pd surface energy anisotropy and oxygen adsorption).

– The interpretation of the results in terms of the surface energy anisotropy of Pd appropriately builds on the framework originally proposed in Ref. 23. Could this be more directly related to the classical Wulff construction/equilibrium shape description?

Response: We have revised the discussion to briefly relate the weighted-curvature description to the Winterbottom construction, which is the supported-film counterpart of the classical Wulff construction. This addition provides a clearer conceptual connection without altering our original analysis (see Results section: Effects of Pd surface energy anisotropy and oxygen adsorption).

Reviewer #3 (Remarks to the Author):

In this paper, the authors have prepared the single crystalline Pd thin films and transferred on amorphous SiO₂ substrate for the investigation of solid state dewetting behavior. They have also changed the annealing gas conditions with different Ar and H₂ amounts and mixtures. Interesting structures evolutions have been observed. The authors have also tried to trace the dewetting process and behavior via numerical methods. As the authors mentioned, the

dewetting of single crystalline metal films have been already studied, and the big difference in this paper is to use amorphous substrate instead of single crystalline MgO substrates. The Results are interesting, but the paper is not clearly formulated, and the novelty is also not high enough for NC. I would suggest to reject and transferred into another journal. Here are some comments:

Response: We appreciate the reviewer's careful consideration.

First, we would like to clarify the primary advances of this work.

As noted by the reviewer, patterned solid-state dewetting of single-crystal metal films has been previously studied, primarily on single-crystal MgO substrates. A central contribution of the present study is that we enable deterministic and orientation-dependent pattern formation on amorphous substrates by combining multilayer epitaxial growth with a film-transfer process. This removes the dependence on MgO substrates, whose use is limited by the need for long-duration preheating and highly sensitive growth conditions. In contrast, our method allows epitaxial Pd films to be grown at room temperature on Si using a standard e-beam evaporator and subsequently transferred onto amorphous SiO₂, greatly expanding the accessibility of this patterning approach.

A second contribution is the systematic identification of the physical and chemical origins governing the ambient-dependent pattern morphologies through a multiscale approach that integrates MD and DFT calculations with KMC simulations. Prior studies did not incorporate adsorbate binding energies or vacancy formation energies and therefore could not account for the effects of annealing ambient or adsorbate-induced anisotropy. By including these components, we provide mechanistic explanations for both the orientation dependence and the ambient dependence of the observed pattern formation—features that were not addressed in earlier works.

We have revised the manuscript for clarity and organization following the reviewer's comments, and detailed responses to individual points are provided below.

(1) It is confused in your statement that oxygen and hydrogen chemisorption favored the formation of <011> in-plane facets in the Pd(100) films. Because you have just described that formation of primary <001> in-plane facets only with small amount of H₂.

Response: We agree that the earlier description may have caused confusion, and we have clarified this point in the revised manuscript. Since thermal annealing was carried out at atmospheric pressure, the Pd surface becomes oxidized in the absence of a reducing gas. Hydrogen was introduced primarily to prevent oxidation or to reduce the surface oxide. At a low flow rate such as 0.5 sccm, a substantial fraction of the supplied hydrogen is consumed during oxide reduction, leaving only a limited amount available for adsorption. Under these conditions, adsorbate-induced changes in surface energy anisotropy are insufficient, and the primary in-plane facets remain aligned along the <001> directions.

When the hydrogen flow rate is increased to 3–5 sccm, additional hydrogen remains available after the reduction step, which may allow hydrogen adsorption to influence the surface energy

anisotropy. In this regime, primary $\langle 011 \rangle$ in-plane facets can appear. This clarification has been added to the revised manuscript

In the calculation suggested by another reviewer, however, the H coverage at 800°C was found to be too small to solely account for the reemergence of the dominant $\langle 011 \rangle$ facets. Hence, we noted that further investigation (e.g., in-situ surface characterization) is needed to clarify the origin underlying the reemergence at high H₂ flow rates (see Results section: Effects of Pd surface energy anisotropy and oxygen adsorption and DFT-calculated adsorption energetics and their implications for dewetting).

(2) In figure 3, you said there is no Si in the Pd structure confirmed by SAED. However, it is better to confirm it by investigation with EDX or similar method.

Response: We agree that confirming the absence of Si using EDS is appropriate. We performed EDS analyses on one of the patterns shown in Fig. 3(d). The results, including two line scans acquired at different positions of the pattern, are now provided in the Supplementary Information (Fig. S6). As shown in Fig. S6, the Si signal within the Pd pattern is indistinguishable from the background, indicating that Si is not incorporated into the Pd structure (see Results section: Effects of Pd surface energy anisotropy and oxygen adsorption).

(3) Your study is about the investigation on dewetting of the single crystalline Pd on the amorphous SiO₂ substrate, and suggests that surface energy anisotropy of films and gas adsorption are more important for the structure evolution than the film-substrate interface energy. But how about the influence of strain energy? The thermal expansion coefficients between metals and oxides are large.

Response: We compared the reductions of surface/interfacial energy and strain energy following the approach used in previous studies (ref. 2 in the Supplementary Information). The detailed analysis is provided in Note S3 of the Supplementary Information.

As the reviewer pointed out, the thermal expansion mismatch between Pd and SiO₂ generates thermal stress. The purely elastic thermal stress at 800 °C is approximately 1.72 GPa; however, since the samples were heated to 800 °C over ~22 min, the actual stress at the onset of dewetting is expected to be significantly lower than this theoretical value. For this reason, we used the yield stress of Pd as the relevant stress to estimate the strain-energy reduction.

The yield stress of Pd depends strongly on microstructure, and values reported in the literature vary widely across different specimen types. To cover a broad and conservative range, we evaluated the strain energy using several reported yield-stress values measured at room temperature for different Pd specimens, including nanocrystalline Pd films (~500 MPa), cold-worked Pd (~250 MPa), and annealed Pd (~40 MPa). Among these, nanocrystalline Pd films show the highest reported value (~500 MPa), which we used as a conservative upper bound.

Because this value (~500 MPa) corresponds to nanocrystalline Pd, it represents a conservative upper bound. Epitaxial Pd(100) films—lacking grain-boundary strengthening—and the elevated annealing temperature are both expected to substantially reduce the actual yield stress relative to this upper-bound estimate. Using this upper-bound stress together with a particle-

radius-to-film-thickness ratio of ~ 4 (estimated from ref. 1 in the manuscript), the reduction in surface/interfacial energy was calculated to exceed that of strain energy by a factor of approximately 12.5. When lower yield-stress values reported for cold-worked and annealed Pd are used, the dominance of surface/interfacial energy becomes even more pronounced (ratios of ~ 50 and ~ 1960 , respectively).

These results show that surface and interfacial energies dominate the driving force for dewetting under our experimental conditions. Nonetheless, we acknowledge that stress may still influence certain aspects of the evolution—for example, by lowering the hole-nucleation barrier or affecting the roughening temperature. These points are noted in Note S3 of the Supplementary Information.

(4) As you mentioned that the interface energy is not important for the structure evolution due to the amorphous nature, but it is strange that in figure 4b, the structures are clearly different with different tilt angles. Why? Influence of gas flow direction?

The reviewer's confusion likely arises from distinguishing the role of the substrate from the role of the initial in-plane crystallographic orientation of the Pd pattern. Because the SiO_2 substrate is amorphous, the film–substrate interface does not impose any preferred in-plane alignment. Thus, the interface energy does not vary with the in-plane rotation of the pattern.

However, the film itself is single-crystalline, and therefore its surface energy anisotropy depends on the crystallographic orientation of the initial edges. Dewetting proceeds by edge retraction processes that depend on the relative stability of $\langle 001 \rangle$ and $\langle 011 \rangle$ -type edges, which is why patches with different in-plane orientations evolve into different morphologies even on an amorphous substrate.

The patterns in Figs. 4(b) and 4(c) correspond directly to the experimental cases in Figs. 2(a) and 2(b), where only the initial crystallographic edge orientations differ. The differences in the resulting morphologies therefore originate from the intrinsic surface energy anisotropy of Pd and are not related to the substrate or the gas-flow direction.

(5) The whole arguments about the influence of anisotropy of surface energy and gas adsorption are also not clear enough. By anisotropy of surface energy you mean the orientation dependent aspect. But surface energy is the energy associated with two phases (solid and gas), and this means if you have different gas or gas mixture, then the interface energy is different, and this can also lead to different anisotropy of surface energy.

We agree that different gas environments can influence the solid–gas interfacial energy, and this is one of the central points addressed in our work. In our study, the ambient-dependent morphological evolution was interpreted in terms of facet-dependent chemisorption, where adsorbates bind with different strengths to Pd surfaces of different crystallographic orientations. These facet-specific binding energies modify the relative surface energies of the facets and therefore alter the effective surface-energy anisotropy.

This concept has already been discussed in the manuscript, where the DFT-calculated binding energies for each facet were used to explain the experimentally observed changes in facet

stability and pattern evolution (see Results section: Effects of Pd surface energy anisotropy and oxygen adsorption and DFT-calculated adsorption energetics and their implications for dewetting).

Responses to reviewer comments

We sincerely appreciate the reviewers' comments and recommendations for publication. Our responses are provided below.

Author's Note to Editor and Reviewers

There were typographical errors in the equation used to estimate the adsorbate coverage. The E_0 has been corrected to $2E_0$, and the notation was further revised to $2BE(O^*)$ for consistency with the other equations in the manuscript. The coverage values reported in the manuscript were originally calculated using this formulation.

$$f_0 = \frac{\sqrt{\exp(\mu_{O_2} - 2BE(O^*))/kT}}{1 + \sqrt{\exp(\mu_{O_2} - 2BE(O^*))/kT}}$$

Minor numerical adjustments to the adsorption coverage data were also made following a refined treatment of the gas-phase enthalpic contribution to the gas-phase chemical potential. As a result, three numerical values in the following sentences have been updated. However, these changes do not affect the discussion and conclusions of the manuscript.

(original)

The surface-energy ratio between the (100) and (111) surfaces was calculated to be ~ 1.119 , which corresponds to $\zeta = 7.5$, as the O coverages were ~ 0.0227 and ~ 0.0561 ML on (100) and (111) surfaces at an oxygen partial pressure of $\sim 2.91 \times 10^{-3}$ atm at 800 °C, respectively.

(revised)

The surface-energy ratio between the (100) and (111) surfaces was calculated to be ~ 1.119 , which corresponds to $\zeta = 7.5$, as the O coverages were ~ 0.0227 and ~ 0.0561 ML on (100) and (111) surfaces at an oxygen partial pressure of $\sim 1.10 \times 10^{-3}$ atm at 800 °C, respectively.

(original)

Using an upper-bound estimate based on an Ar–H₂ ambient, increasing the H₂ flow rate to 3 and 5 sccm in 50 sccm Ar corresponds to partial pressures of ~ 0.0566 and ~ 0.0909 atm, which yield ζ values of ~ 2.50 and ~ 2.57 , respectively.

(revised)

Using an upper-bound estimate based on an Ar–H₂ ambient, increasing the H₂ flow rate to 3 and 5 sccm in 50 sccm Ar corresponds to partial pressures of ~ 0.0566 and ~ 0.0909 atm, which yield ζ values of ~ 2.668 and ~ 2.798 , respectively.

Reviewer #1 (Remarks to the Author):

The authors have thoroughly addressed all of my concerns, and I still recommend publication in Nature Communications. I read the other reviewers concern that the results are not convincing enough for publication in Nature Communications. I think it is true that any computational attempt at predicting the behaviour of surface energies and kinetics is fraught. Surfaces are complex and one can (and should!) always worry that something has been left out (as I stated in my review). But that does not diminish the importance of work such as presented here that makes a serious attempt using all the tools available to account for complex experimental observations. If the presented model of dewetting turns out to still be incomplete, it will be for an interesting reason.

Response: We thank the reviewer for the support of our work and recommendation for publication.

Reviewer #2 (Remarks to the Author):

The authors carefully revised their manuscript, addressing most of the comments raised in my previous report, clarifying crucial points, and providing additional evidence. The "stronger case" I was referring to has now been made.

I acknowledge the relevance of making the self-assembly of nanostructures exploiting dewetting compatible with relatively standard processes (room-temperature epitaxial growth, avoiding state-of-the-art vacuum conditions). The proposed analysis of chemical effects in the gas phase, which is deeper than what has been reported in the literature so far, is also a relevant aspect.

I believe the authors have appropriately downplayed the claim of a fully predictive modeling framework, and I agree that this work represents "a step toward a more predictive modeling framework," although further efforts and understanding (now also discussed in the manuscript) are still needed. It remains, however, that the title points to "predictive patterning of ..." rather than to the aspects mentioned above ("transferred" is relatively unclear at first reading), and that the predictive and reproducible character of dewetting exploiting pre-patterning has already been discussed extensively in the literature (as properly acknowledged by the authors). I therefore suggest making the actual relevant element of novelty more clearly evident from the very beginning (e.g., in the title).

I thank the authors for the extensive explanations concerning KMC simulations and appreciate the effort made to clarify their claims, also through additional discussion in the manuscript. The point raised regarding the proposed "guidelines" is now fully clarified as they indeed refer to the specific process considered here (including ambient conditions) rather than to controlled solid-state dewetting in general. The same applies to remarks on the relevance of the transfer step.

Importantly, quantitative analyses have been added (new Fig. 6). Moreover, the extended simulation campaign, which increases the reliability of the claims, is definitely a plus and

demonstrates the robustness of the process, which was not clearly conveyed in the previous draft.

I believe the manuscript can now be accepted for publication, while noting that further tailoring the visibility of the key messages to better convey the actual points of novelty could further increase the work's impact and accessibility.

Response: We thank the reviewer for the positive evaluation of the revised manuscript and the suggestion. We revised the first sentence of the Abstract to clarify that, although specific dewetting patterns have been demonstrated previously, extending them toward the fabrication of functional structures requires improved predictability and extensibility. We also added a sentence in the Introduction explaining that film transfer removes the need for single-crystal oxide substrates and associated process constraints.

Reviewer #3 (Remarks to the Author):

it is great that the authors have made big efforts to revise the manuscript, and the revisions are properly done. I suggest to accept it for publication. But the language should be polished again. For example, the sentence in the abstract is hard to understand: “The anisotropies of Pd surface energy and gas adsorption strength were found to cause the <001> and <011> in-plane facets to form and favor the latter as oxygen adsorption increases or hydrogen flow rate is sufficiently high.”

Response: We thank the reviewer for the recommendation for publication and the suggestion regarding language polishing. We have revised several sentences to improve clarity. For example, the sentence mentioned by the reviewer has been revised as follows:

(original) The anisotropies of Pd surface energy and gas adsorption strength were found to cause the <001> and <011> in-plane facets to form and favor the latter as oxygen adsorption increases or hydrogen flow rate is sufficiently high.

(revised) The anisotropies of Pd surface energy and gas adsorption strength lead to <001> and <011> in-plane facets, favoring the latter as oxygen adsorption increases or the hydrogen flow rate is sufficiently high.